# How Abilities in Large Language Models are Affected by Supervised Fine-tuning Data Composition

## Abstract

Large language models (LLMs) with enormous pre-training tokens and parameter amounts emerge abilities, including math reasoning, code generation, and instruction following. These abilities are further enhanced by supervised fine-tuning (SFT). The open-source community has studied on ad-hoc SFT for each ability, while proprietary LLMs are versatile for all abilities. It is important to investigate how to unlock them with multiple abilities via SFT. In this study, we specifically focus on the data composition between mathematical reasoning, code generation, and general human-aligning abilities during SFT. From a scaling perspective, we investigate the relationship between model abilities and various factors including data amounts, data composition ratio, model parameters, and SFT strategies. Our experiments reveal that different abilities exhibit different scaling patterns, and larger models generally show superior performance with the same amount of data. Mathematical reasoning and code generation improve as data amounts increase consistently, while the general ability is enhanced with about a thousand samples and improves slowly. We find data composition results in various abilities improvements with low data amounts, while conflicts of abilities with high data amounts. Our experiments further show that composition data amount impacts performance, while the influence of composition ratio is insignificant. Regarding the SFT strategies, we evaluate sequential learning multiple abilities are prone to catastrophic forgetting. Our proposed Dual-stage Mixed Fine-tuning (DMT) strategy learns specialized abilities first and then learns general abilities with a small amount of specialized data to prevent forgetting, offering a promising solution to learn multiple abilities with different scaling patterns.

## 1 Introduction

Recent research has demonstrated the remarkable and versatile proficiency of large language models (LLMs) in dealing with a variety of real-world tasks expressed in natural languages (Ouyang et al., 2022a; Anil et al., 2023; OpenAI, 2023). Among the tasks, LLMs especially emerge with three outstanding abilities in reasoning (Cobbe et al., 2021; Wei et al., 2022), coding (Chen et al., 2021), and aligning general human intentions (Ouyang et al., 2022a), which have drawn much attention from the LLM research community. In order to further incentivize such abilities, it necessitates supervised fine-tuning (SFT) stages on annotated task data. However, existing research has mostly conducted separate SFT investigations on each of the three tasks, where reasoning and coding abilities require SFT on in-domain human-annotated or augmented data (Yuan et al., 2023b; Luo et al., 2023) while diverse and complex human instructions are applauded for aligning human intentions (Wang et al., 2023c; Taori et al., 2023; Xu et al., 2023; Zhou et al., 2023; Wang et al., 2023a; Lu et al., 2023). As shown by the strong performance of proprietary LLMs such as GPT-4 (OpenAI, 2023) and Claude, LLMs have the potential to master all the tasks in one model. Therefore, it is of paramount importance to investigate the versatile performance of SFT with composite task data, and understanding and addressing the challenges posed by the data composition problem in the SFT stage is crucial for further enhancing the capabilities of LLMs in a comprehensive manner.

In essence, the tasks of reasoning, coding, and aligning human intentions are of different characteristics. Reasoning and coding tasks require ad-hoc abilities of complex and detailed logic in

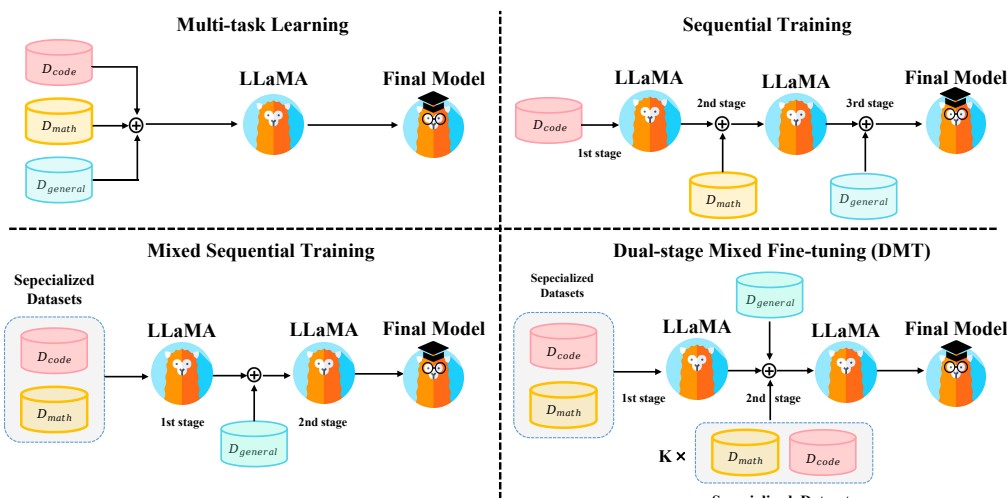

Figure 1: The illustration of four different training strategies in this paper.

decomposing task instructions and dealing with non-linguistic and symbolic features (Chen et al., 2021; Huang & Chang, 2023), whereas aligning human intentions requires versatility and understanding obscure intentions expressed in human instructions (Lu et al., 2023). Given the fundamental difference among the tasks, multi-task learning with composite data fine-tuning for small-scaled pre-trained language models is prone to catastrophic forgetting (De Lange et al., 2022), hindering the fine-tuned performance of one model on separate tasks. Many efforts have been made to compensate for the phenomenon (Liang et al., 2021; Xu et al., 2021; Yuan et al., 2023a). There has also been research discovering that scaling up the pre-trained language model scale and the fine-tuning data scale are beneficial for zero-shot out-of-domain generalization on various linguistic tasks while leaving out the assessment of in-domain performance (Sanh et al., 2022; Chung et al., 2022a; Longpre et al., 2023). Given the increased capacity of LLMs, the multi-task performance by SFT on composite data of essentially different downstream tasks is less studied. Understanding the SFT performance with composite data and corresponding scaling patterns is of great utility in practice.

In this study, we focus on the data composition problem among **mathematical reasoning**, **code generation**, and **general human-aligning abilities** in SFT. We aim to comprehensively investigate the relationship between model performance and different factors including data amount, data composition ratio, model scales, and SFT training strategies. We also investigate how the relationship varies under different scales. Specifically, we focus on the following four research questions:

1. *How do math reasoning, coding, and general abilities scale with SFT data amounts?*

2. *Are there performance conflicts when combining these three abilities in SFT?*

3. *What are the key factors that induce the performance conflicts?*

4. *What are the impacts of different SFT strategies for composite data?*

To answer these questions, we conduct experiments on three benchmarks, which are GSM8K (Cobbe et al., 2021) for mathematical reasoning, HumanEval (Chen et al., 2021) for coding, and MT-Bench (Zheng et al., 2023) for general human alignment. We fine-tune LLMs on the related training data to activate these abilities. Furthermore, we conduct extensive analysis regarding model parameter scales ranging from LLaMA 7B to 33B (Touvron et al., 2023) and explore four different SFT strategies shown in Figure 1: multi-task learning, sequential training, mixed sequential training, and dual-stage mixing fine-tuning (DMT), providing empirical guidance for learning a versatile LLM with composite SFT. The key findings of this paper can be summarized as follows:

- Different SFT abilities exhibit distinct scaling patterns, while larger models show better performances with the same data amount generally.

- Compared to single ability learning, multi-task learning multiple abilities exhibits improvement in low-resource and decline in high-resource. Additionally, as the model size increases, there is a greater performance gain in low-resource settings for math and general abilities.

- Data amounts directly influence each ability, while the data ratio is insignificant.

- Multi-task learning lead to conflicts, while sequential training results in catastrophic forgetting. Our proposed DMT effectively alleviates both performance conflicts and catastrophic forgetting in the SFT phrase, achieving a balance between general and specialized abilities.

## 2 RELATED WORKS

**Supervised fine-tuning in Large Language Models** Large language models (LLMs) undergo the SFT stage to further unlock the performance in task solving and aligning human instruction. We slightly abuse the term SFT to refer to general sequence-to-sequence fine-tuning, including but not limited to SFT for human alignment, instruction fine-tuning, and downstream task fine-tuning. Recent research explored multi-task instruction fine-tuning of pre-trained LLMs to enable better zero-shot performance on various downstream NLP tasks (Sanh et al., 2022). (Chung et al., 2022a; Longpre et al., 2023) attempted to exhaust existing NLP tasks and curated a massive dataset, FLAN, for instruction fine-tuning. Open-sourced (Chung et al., 2022b) and proprietary LLMs (Singhal et al., 2022) fine-tuned on FLAN exhibited improved zero-shot downstream performance on various held-out NLP tasks. However, the influence of multi-task training of LLMs on in-domain performance is less studied. With the success of proprietary LLMs, especially ChatGPT, there has been increasing attention on SFT to align LLMs to human intentions (Ouyang et al., 2022b). Instead of generating SFT data from crowd-resourcing, recent research explored to generate data from proprietary LLM user logs (Chiang et al., 2023; Wang et al., 2023a), prompting proprietary LLM (Wang et al., 2023c; Taori et al., 2023; Lei et al., 2023; Xu et al., 2023). Various analyses and methods have also been proposed to increase the SFT data quality (Zhou et al., 2023; Wang et al., 2023b; Lu et al., 2023) to achieve better alignment of open-resourced LLMs with humans. Besides, LLMs can also benefit from SFT for mathematical reasoning (Cobbe et al., 2021; Hendrycks et al., 2021; Yuan et al., 2023b; Yue et al., 2023) and code generation tasks (Chaudhary, 2023; Luo et al., 2023).

**Scaling Laws in Large Language Models** The exceptional performance of LLMs comes from scaling up model sizes, data amounts, and computational costs to massive scales. Therefore, it is crucial to explore the model performance across an exponential range of scales. Many endeavors have been made to discuss the scaling laws for pre-training (Anil et al., 2023; Hoffmann et al., 2022), transfer learning (Chronopoulou et al., 2019), preference modeling (Gao et al., 2022) and mathematical reasoning (Yuan et al., 2023b). In this paper, we also explore the SFT performance with composite data from the perspective of different scales of model sizes and data amounts.

## 3 EXPERIMENTS

We have SFT datasets $\{D_1, D_2, ..., D_k\}$ where each $D_i = \{q_{i,j}, r_{i,j}\}_j$ contains queries and responses from one source. We consider each SFT dataset to correspond to one ability and we also have $k$ in-domain metrics to measure them. We investigate the performances of in-domain metrics with different dataset compositions ($D \subset \cup_{1 \le i \le k} D_i$) and training strategies on different sizes of LLMs.

### 3.1 EXPERIMENT SETUP

We collect three SFT datasets $\{D_1, D_2, D_3\}$ including GSM8K RFT (Yuan et al., 2023b), Code Alpaca (Chaudhary, 2023), and ShareGPT (Chiang et al., 2023) to represent math reasoning, coding, and general human-aligning ability SFT dataset respectively. We will integrate a new SFT dataset $D$ by these three datasets to investigate how data composition affects the model performances. We use GSM8K test set (Cobbe et al., 2021), HumanEval (Chen et al., 2021), and MT-Bench (Zheng et al., 2023) to measure abilities including math reasoning, coding, and general human-aligning. We use LLaMA (Touvron et al., 2023) series as our pretrained language models and use FastChat framework (Zheng et al., 2023) for fine-tuning. We fine-tune models with 3 epochs and a peak of 2e-5 learning rate. The batch size during SFT is 16. More details about SFT datasets, evaluation metrics and implementations can be found in Appendix A, B and C.

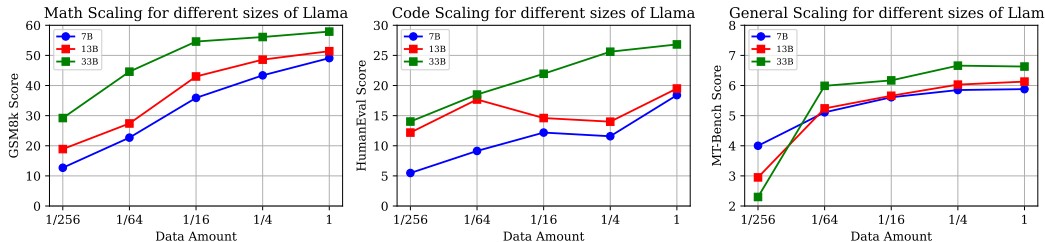

Figure 2: The scaling curve of different sizes of LLaMA in three individual domains.

## 3.2 RQ1. INDIVIDUAL ABILITY PERFORMANCE VS. DATA AMOUNT

The instruction following ability can be activated via SFT on datasets like ShareGPT which contain around 100 thousand samples. However, Zhou et al. (2023) demonstrates that strong base models can achieve human alignment with just 1000 samples. Specialized abilities such as math reasoning require a large amount of data (Cobbe et al., 2021; Yuan et al., 2023b), unlike general abilities. Therefore, it is crucial to investigate how each ability improves as the data amount increases.

**Experimental Design:** We conduct SFT on LLaMA of various sizes using {1, 1/4, 1/16, 1/64, 1/256} proportions of the training set obtained from GSM8K RFT, Code Alpaca, and ShareGPT seperately. This allowed us to evaluate each ability with various data sizes and model sizes.

**Results and Analysis.** Figure 2 shows the individual data scaling curves for different abilities after SFT. We find that: **Different abilities exhibit different scaling curves.** To be more specific, mathematical reasoning capability shows a positive correlation with the data amount across various model sizes which is consistent with Yuan et al. (2023b). Similarly, general human-aligning ability demonstrates an almost monotonically increasing scaling curve. However, it is noteworthy that general ability emerges with only around 1k data samples (ranging from 1/256 to 1/64), and after reaching a certain threshold (1/64), their performances improve slowly. This further supports Zhou et al. (2023), indicating that a small amount of high-quality SFT data is possible for the emergence of general human-aligning ability in LLMs. On the other hand, code ability exhibits an irregular scaling curve when the model's parameter count is small (7B & 13B). However, when the parameter count increases to 33B, its coding performance shows an approximately log-linear trend with the data amount. One possible explanation is that Code Alpaca and the samples in HumanEval have different distributions. Larger models can capture shared knowledge across code data distributions in the in-domain samples, which enables them to exhibit some level of generalization to out-of-distribution (OOD) samples. Another observation is **larger models show better performances with the same data amount generally.** The outlier is with very little data (1/256), smaller models may outperform larger models. If there is enough data, larger models have stable better performances.

## 3.3 RQ2. PERFORMANCE DIFFERENCE VS. MIXED DATA AMOUNT

We should deliver a versatile model that requires us to mix various SFT datasets and apply SFT. We want to ask how each ability varies due to SFT dataset mixtures. We investigate it with different amounts of mixed data and compare them with individual ability performance.

**Experimental Design:** For the individual source setting, consistent with the setup in RQ1, we performed fine-tuning on LLaMA models of different sizes using {1, 1/4, 1/16, 1/64, 1/256} amounts of training data from GSM8K, Code Alpaca, and ShareGPT separately. For the mixed source setting, we sampled {1, 1/4, 1/16, 1/64, 1/256} amounts of training data from GSM8K, Code Alpaca, and ShareGPT, and directly mixed them according to the corresponding proportions. In this way, we constructed datasets with fixed proportions of different ability domains, while varying the total data amount. These datasets are then used for fine-tuning the LLaMA models.

**Results and Analysis.** Figure 3 presents results of LLaMA of different sizes on three benchmarks under the individual source and mixed source settings. The following observations are made: **Abilities are improved with low-resource and are decreased with high-resource compared to individual source abilities.** In the case of LLaMA-7B, compared to the data scaling curve of the individual

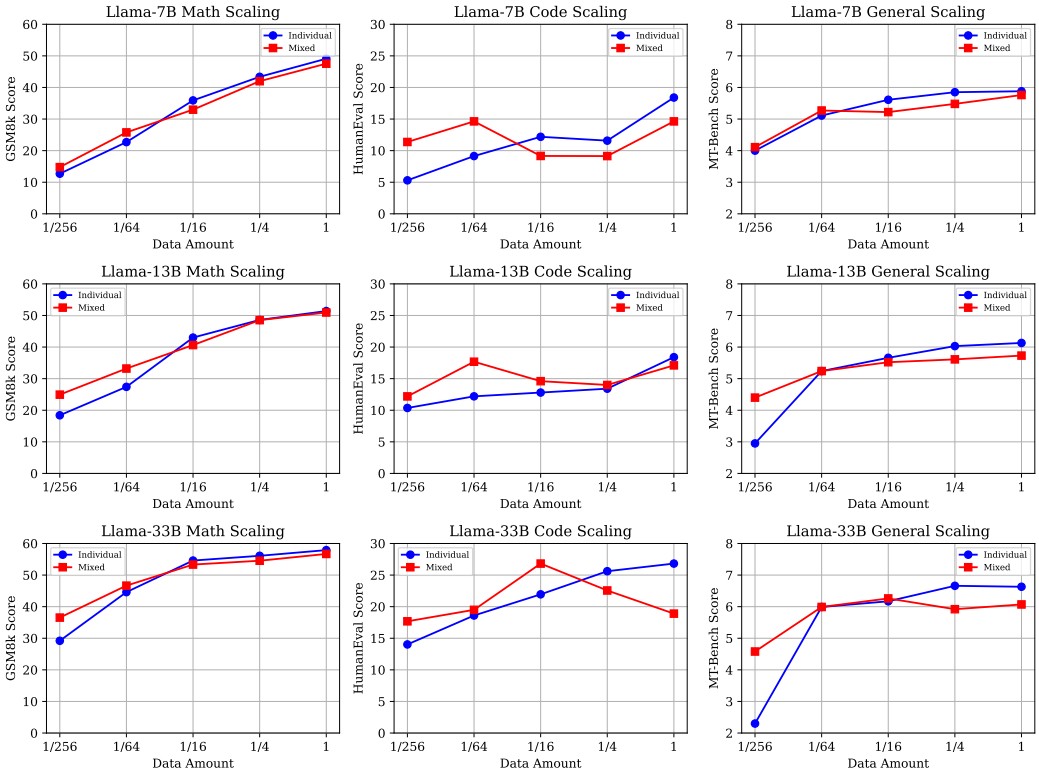

Figure 3: Comparative experiments between mix domains and individual domains for LLaMA.

source setting, the models fine-tuned with mixed source data consistently demonstrated performance conflicts among the three ability domains at high resources (100%). However, as the data volume decreased, a turning point in performance is observed between the two settings in the data range of 1/64 to 1/16. Notably, the models fine-tuned with mixed source data exhibited performance gains at low resources (1/256), indicating that SFT data from different sources benefit each other in a low-resource setting. However, when there is enough data, data from other sources could be viewed as noise for in-domain generalization. **As the model size increases, the performance gain in low-resource settings also increases for math and general abilities.** In the case of the 13B and 33B models, it is obvious that the scaling curve for the mix source setting follows a similar trend observed in previous analyses, with the presence of performance intersection points as the data volume scales. However, a crucial distinction arises, whereby larger models exhibit more pronounced performance gains under low resources as the size of model parameters increases. The outlier is the LLaMA-7B (code only, 1/256). A possible reason is the introduction of a small amount of unseen code data easily disrupts the original code ability of the pretrained model, as supported by its low HumanEval score (less than 6). In conclusion, our finding implies that larger language models excel in acquiring general and specialized abilities from diverse data sources under low-resource conditions.

### 3.4 RQ3. PERFORMANCE DIFFERENCE VS. DATA COMPOSITION RATIO

We observe ability conflicts under high-resource settings, and we want to investigate the reason causes conflicts. Two possible factors are the **data amount** of other abilities is too high or the **data ratio** of other abilities is too high. Here we conduct experiments to investigate the data ratio factor.

**Experimental Design:** We consider coding and mathematics as a combined specialized data source, and the ShareGPT as the general data source. We designed three setups as follows which control the amount of one source of data and vary the ratio between general and specialized data.:

**1. Fixed general data, scaling specialized data:** We use a full training set of ShareGPT and sampled different proportions {1, 1/4, 1/16, 1/64, 1/256} of GSM8K RFT and Code Alpaca as a mixture.

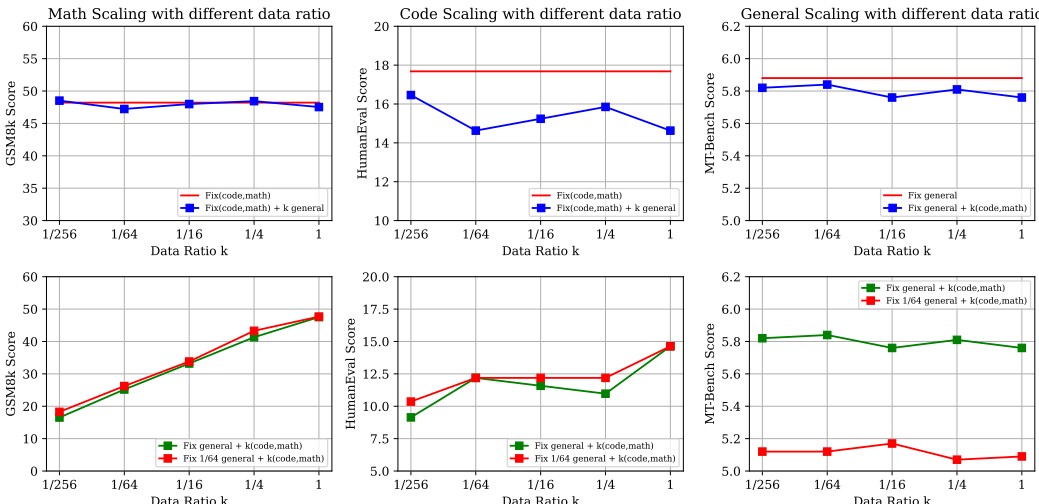

Figure 4: Different data ratio (k) between specific abilities and general abilities on three benchmarks.

**2. Fixed specialized data, scaling general data:** We use a full training set of GSM8K RFT and Code Alpaca and sample different proportions of ShareGPT as a mixture.

**3. Fixed 1/64 general data, scaling specialized data**: Motivated by LIMA's setup (Zhou et al., 2023), we used a 1/64 ShareGPT set (about 1500 examples) and sampled different proportions of GSM8K RFT and Code Alpaca as a mixture.

**Results and Analysis.   Does the performance of the model vary with different ratios of general and specialized data?** As illustrated in the top three graphs of Figure 3, we conduct ablation studies of the data ratio ($k$) between specialized and general abilities. To be noticed ratio is normalized by data amount, for example, $k = 1$ means $\frac{\text{specialized use data amount}}{\text{general use data amount}} = \frac{\text{specialized all data amount}}{\text{general all data amount}}$. We utilize a fixed specialized data setting (directly mixing 100% code & math data for training) and a fixed general data setting (100% general data for training) as the baseline and observe: (1) With the increase in the ratio of general data from 1/256 to 1/1, *Fixed specialized data, scaling general data* setup exhibits similar performance to the setup that *Fixed specialized abilities* in terms of math reasoning. This suggests that variations in the data ratio $k$ have minimal impact on math ability. We consider the reason that math and general abilities are non-conflict since they are too different in the semantic space. However, when considering HumanEval, the *Fixed specialized data, scaling general data* setup displays noticeable fluctuations compared to the baseline. We attribute this to the inclusion of a certain proportion of code data in ShareGPT. Due to the differences in data format and distribution, the presence of similar data features exacerbates the performance conflicts between abilities when the data ratio $k$ increases. Further analysis of the distribution of different abilities is discussed in Section 4.1. (2) With the increase in the ratio of specialized data from 1/256 to 1/1, the setup that *Fixed general data, scaling specialized data* displayed no significant performance changes compared to the baseline. This echoes our hypothesis that when **there are significant differences in task formats and data distributions between different SFT abilities, the impact of data ratio is minimal**. However, when **there is some degree of similarities, the data ratio can lead to noticeable performance fluctuations.**

**Under extremely limited general data resources, does the ratio of specialized data have an impact on the model's performance?** We further explore the impact of different ratios of specialized data when the model has just acquired a certain level of general human-aligning ability ($k = 1/64$). The bottom 3 graphs of Figure 4 present comparative experiments between two settings. We observe that regardless of whether the data amount for general capabilities is abundant ($k = 1$) or scarce ($k = 1/64$), the performance on MT-Bench shows no significant fluctuations with varying proportions of specialized data. Furthermore, in mathematical reasoning, 1/64 general data setup exhibited a scaling trend that is almost identical to the full general data setup. However, for coding ability, with

the same amount of code data and different ratios, code abilities are different in the two settings. We still consider the reason is code data are partly related to ShareGPT data and cause the performance difference and provide an analysis in Discussion 4.2.

## 3.5 RQ4. PERFORMANCE DIFFERENCE VS. TRAINING STRATEGY

We could feed these SFT datasets into models with different training strategies. In this section, We experiment with these settings and investigate how they influence each ability's performance.

**Experimental Design:** Firstly, we introduce three kinds of naive training strategies as follows:

**1. Multi-task learning:** We directly mix different SFT data sources $D = \cup_{1 \leq i \leq k} D_i$ and applying SFT. If we view each data source as a different task, this can be viewed as multi-task learning.

**2. Sequential Training:** We sequentially apply SFT on each dataset. Specifically, we sequentially trained on coding, math reasoning, and the general ability dataset. Since the general ability is the most important one for human alignment, we put ShareGPT as our last dataset.

**3. Mixed Sequential Training:** We apply multi-task learning on specialized datasets(code, math) first and apply SFT on the general ability dataset. These three approaches are presented in Figure 1.

**Results and Analysis:** Table 15 presents performances under different training strategies in terms of mathematical reasoning, code generation, and general human-aligning ability. Multi-task learning preserves specialized abilities among these strategies while hurting the general ability most among them. Sequential training and mixed sequential training preserve general ability while losing too many specialized abilities. The observed outcome is in accordance with our expectations, as during the final fine-tuning phase, the mixed sequential training strategy remains unaffected by specialized data sources, thereby effectively preserving its generalization capability. However, an inherent drawback of multi-stage training is the occurrence of catastrophic forgetting of prior knowledge, which motivates us to further explore methods that can alleviate catastrophic forgetting of specialized abilities while maximizing the preservation of general capability.

**4. Dual-stage Mixed Fine-tuning (DMT):** Based on our observation from RQ1 to RQ4, we propose a new training strategy that can reduce the ability conflict during multi-task learning and relieve the issue of catastrophic forgetting during sequential training. From RQ1, the model needs large data amounts to activate specialized abilities. From RQ2, multi-task learning with all amounts of specialized data and general data will hurt each ability. From RQ3, a small amount of specialized data will not affect the general ability performance. From RQ4, (mixed) sequential training forgets specialized abilities. So the model needs to learn large amounts of specialized data and should not forget them during learning general ability. A natural choice is to learn full amounts of specialized data first and add a small amount of specialized data to general data during the last stage of sequential training to prevent forgetting. As shown in Figure 1, we first apply SFT on the specialized dataset which is same as the first stage of the mixed sequential training strategy. For the second stage, we perform SFT with a mixed data source comprising a combination of the general data and varying proportions $k$ (1, 1/2, 1/4, 1/8, 1/16, 1/32) of code and math data. Adding code and math data in the second stage helps models to recall the specialized ability. The results of DMT ($k = 1/256$) are presented in Table 2 and the detailed scaling analysis of proportion $k$ can be found in the discussion.

**Model Accuracy vs. DMT Strategies.** In Table 15, LLaMA-7B with DMT ($k = 1/256$) strategy perform significant improvement in mathematical reasoning (32.6 to 41.92) and code generation (15.24 to 17.68) compared to the mixed sequential training strategy, which indicates a significant alleviating effect of mixing specialized capability data in the last fine-tuning stage on catastrophic forgetting. Surprisingly, DMT ($k = 1/256$) even exhibits a slight improvement on MT-Bench, further highlighting its ability to alleviate catastrophic forgetting while effectively preserving general capability.

Regarding the 13B and 33B models, DMT ($k = 1/256$) demonstrates noticeable alleviation of catastrophic forgetting in mathematical reasoning (13B: 40.48 to 46.47 / 33B: 44.24 to 56.36) and code generation (13B: 18.3 to 19.5 / 33B: 24.4 to 25.5) compared to the mixed sequential training strategy. Additionally, it significantly retains its general capability (13B: 5.93 to 6.03 / 33B 6.43 to

| Methods | LLaMA -7B | | | LLaMA -13B | | | LLaMA -33B | | |
|---|---|---|---|---|---|---|---|---|---|
| | GSM8K | HumanEval | MT-Bench | GSM8K | HumanEval | MT-Bench | GSM8K | HumanEval | MT-Bench |
| *Individual domain* | | | | | | | | | |
| General only | 11.10 | 10.42 | 5.88 | 14.02 | 16.40 | 6.13 | 26.06 | 24.30 | 6.63 |
| Math only | 49.10 | 6.71 | 2.53 | 51.40 | 12.8 | 2.54 | 57.91 | 15.5 | 3.18 |
| Code only | 4.51 | 18.40 | 4.30 | 5.15 | 17.1 | 3.53 | 6.06 | 26.82 | 4.18 |
| *Different Training Strategies* | | | | | | | | | |
| Multi-task learning | **47.53** | 14.63 | 5.76 | **50.94** | 19.50 | 5.73 | **56.69** | 18.9 | 6.07 |
| Sequential Training | 31.39 | **15.85** | 5.72 | 39.12 | **20.12** | 5.93 | 47.27 | 24.80 | **6.73** |
| Mixed Sequential Training | 32.60 | 15.24 | 6.02 | 40.48 | 18.30 | 5.93 | 44.24 | 24.4 | 6.43 |
| DMT(k=1/256) | 41.92 | 17.68 | **6.08** | 46.47 | 19.50 | **6.03** | 56.36 | **25.00** | 6.69 |

Table 1: The results of LLaMA-7B, 13B, 33B under different training strategies on three benchmarks. The top two results across different strategies are marked with **bold** and underlined.

6.69). Therefore, these results serve as additional validation of the efficacy of DMT in mitigating catastrophic forgetting while maintaining general capability.

# 4 DISCUSSION

## 4.1 VISUALIZATION OF SEMANTIC REPRESENTATION OF DIFFERENT ABILITIES

In the aforementioned analysis of data composition, we observed a significant performance degradation when different data sources are directly mixed. In this section, our aim is to explore the potential mutual influence of semantic representation distributions among different data sources. Specifically, we randomly sampled 100 queries from CodeAlpaca, GSM8k RFT, and ShareGPT datasets and extracted the hidden layer representations located in the 15th layer of the model. Subsequently, we employed the t-SNE toolkit Van der Maaten & Hinton (2008) to visualize the representations of the three types of capabilities. The results in Figure 5 illustrate a notable collapse phenomenon in the semantic representations of both the original LLaMA-13b and LLaMA-13b with DMT (k=1/256). While both models exhibit a certain level of separation in the mathematical data representations, there remains a certain degree of overlap between the representations of code and general samples.

## 4.2 ABLATION OF THE SPECIALIZED DOMAINS IN SHAREGPT

In RQ2, we observe using mixed data sources resulted in improved abilities under low-resource conditions but diminished abilities under high-resource conditions when compared to single data sources. However, the presence of coding and mathematical samples within the ShareGPT introduces uncertainty regarding whether the performance gain under low resources is solely attributed to these specific coding & mathematical data or other orthogonal samples in the general dataset (e.g., translation or extraction). Hence, the objective of this section is to investigate whether the conclusions drawn in Section 3.3 remain valid after removing the code and math samples within ShareGPT.

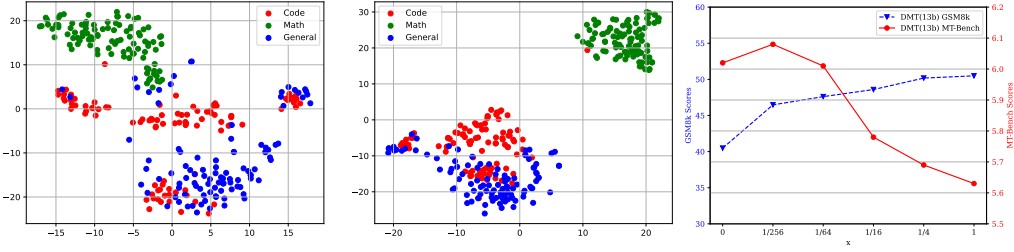

Figure 5: The left two figures show the t-SNE of LLaMA-13B and LLaMA-13B with DMT(k=1/256) stategy. The right figure shows performances of LLaMA-13B with DMT under different $k$.

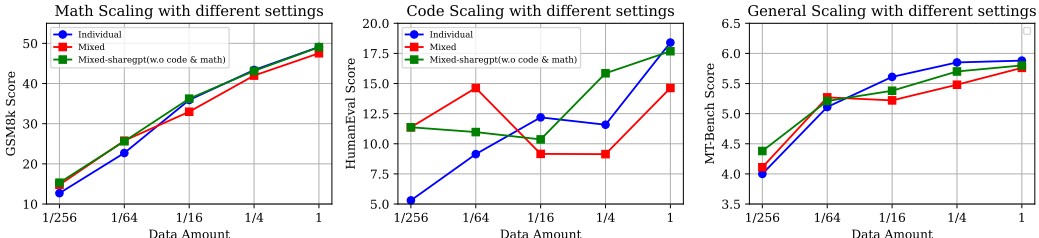

Figure 6: The scaling curve after ablating code and math-related samples from ShareGPT.

**Experimental Design:** We employed an open-set tagger InsTag (Lu et al., 2023) to annotate samples in ShareGPT. To filter out data related to coding and mathematical abilities, we conduct regular expression matching to eliminate instances where the tags contain keywords "code" or "math". Finally, we obtain a ShareGPT dataset devoid of any code or math-related information (reducing from 86K to 63K). In alignment with the settings in Section 3.3, we sampled different proportions of training data (1, 1/4, 1/16, 1/64, 1/256) from GSM8K, Code Alpaca, and the modified ShareGPT dataset (without code math). These samples were directly mixed according to the corresponding proportions. Subsequently, the LLaMA models were fine-tuned by using this mixed dataset.

**Analysis.** Figure 6 shows the results of our experiment. Removing the code and math from ShareGPT not only mitigates the performance conflicts among different abilities to some extent under high-resource conditions but also maintains stable gains in low-resource settings. We propose that the potential reason behind these findings lies in the differences in the distribution of code and math data between ShareGPT, CodeAlpaca, and GSM8K RFT datasets. This distribution gap introduces an extra noise during the SFT phrase, while its removal enables the model to better generalize coding and mathematical abilities. Furthermore, in low-resource scenarios, this phenomenon indicates that the code and math samples in ShareGPT are not the key factor contributing to performance improvements, but rather the diversity and variability of the data (Longpre et al., 2023). In summary, the presence of code math data within ShareGPT does not emerge as a key factor impacting the performance gains identified in Section 3.3, highlighting the generalization of our conclusions.

### 4.3    SPECIALIZED DATA AMOUNT IN DUAL-STAGE MIXING FINE-TUNING

We investigate how different values of $k$ influence model performance and results shown in figure 5. When we adjust $k$ from 0 to 1/256 ($k = 0$ is equal to mixed sequential training), the SFT models show significant improvements in both specialized ability and general human-aligning ability. On the contrary, as $k$ increased from 1/4 to 1, the model exhibited a decline in general ability. We believe this is in line with the findings in RQ2, which concluded that high-resource settings lead to conflicts while low-resource settings lead to gains in mixed sources. Furthermore, as $k$ increased from 1/256 to 1/4, we observe a linear inverse trend between general ability and specialized ability, especially an increase in general ability coincided with a decrease in specialized ability. This suggests $k$ needs to be tuned based on specific requirements in order to achieve a balance between multiple abilities.

## 5    CONCLUSION

We explore the data composition in the SFT phase, focusing on mathematical reasoning, code generation, and general human-aligning abilities. We formulate four research questions to guide our investigation and analyze the scaling trends between different abilities and factors (e.g. data amount, data ratio, model parameters, and training strategies). Our findings reveal distinct scaling patterns among different abilities, with larger models demonstrating superior performance when trained with the same amount of data. Moreover, we observe that mixing data sources in the SFT phase improves performance in low-resource scenarios but diminishes in high-resource scenarios. Interestingly, the phenomenon of low-resource gain becomes more prominent as the model parameter size increases. Furthermore, our observations indicate that data amount directly influences performance conflicts, whereas the impact of data ratio is insignificant within our experimental setup. Finally, regarding the SFT strategies, we demonstrate our proposed DMT strategy effectively alleviates performance conflicts, offering a promising solution to activate multiple abilities.

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

# A    SFT DATASETS

We investigate the data composition issues of mathematical reasoning, coding, and general capabilities in the SFT stage from the following SFT datasets.

- **Code Alpaca** (Chaudhary, 2023) aims to build and share an instruction-following LLaMA model for code generation. which is fully based on Stanford Alpaca and contains 20K data used for fine-tuning the model.
- **GSM8K RFT** (Yuan et al., 2023b) is a mathematical dataset enhanced by integrating multiple reasoning paths based on the original GSM8K dataset (Cobbe et al., 2021) through the rejection sampling. It contains 7.5K questions and 110K responses in the training set.
- **ShareGPT** refers to the multi-turn chatting histories used by Vicuna Chiang et al. (2023). ShareGPT includes 86K human queries and responses from ChatGPT and other chatbots.

The following table presents the statistics of three datasets at different subset proportion (k).

| Data Statistics | GSM8K RFT | Code Alpaca | ShareGPT |
|---|---|---|---|
| K=1/1 | 110142 | 20022 | 86060 |
| K=1/4 | 27535 | 5005 | 21515 |
| K=1/16 | 6883 | 1251 | 5378 |
| K=1/64 | 1720 | 312 | 1344 |
| K=1/256 | 430 | 78 | 336 |

Table 2: Data statistics of three datasets at different subset proportion (k).

# B    EVALUATION METRICS

We use the following metrics to measure the aligned large language models.

- **HumanEval** (Chen et al., 2021) consists of 164 original programming problems, with an average of 9.6 test cases allocated to each problem. To ensure a thorough assessment of the functional correctness of LLM-synthesized code, HumanEval+ extends the number of test cases significantly, averaging at 774.8 test cases per problem. We use the same method as Chen et al. (2021)to obtain unbiased estimates of the pass@k under greedy decoding.
- **GSM8K** (Cobbe et al., 2021) is a math word problem dataset used to measure large language model math reasoning ability. We use the default test set to measure the model. We calculate the score based on greedy decoding accuracy (maj@1).
- **MT-Bench** (Zheng et al., 2023) is a significant benchmark that contribute to the evaluation and advancement of chatbot models and LLMs in different contexts. MT-Bench evaluates LLMs on multi-turn dialogues using comprehensive questions tailored to handling conversations. It provides a comprehensive set of questions specifically designed for assessing the capabilities of models in handling multi-turn dialogues.

We also supplement more benchmark evaluation results in the appendix to verify the generalization of our conclusions:

- **MATH** (Hendrycks et al., 2021) is a dataset with challenging high-school math problems. Problems are classified into the following topics: Prealgebra, Algebra, Number Theory, Counting and Probability, Geometry, Intermediate Algebra, and Precalculus. Problems in MATH are harder and more diverse than in GSM8K. We use 500 test problems from Lightman et al. (2023) as out-of-domain math benchmark.

- **MBPP** (Austin et al., 2021) consists of around 1,000 crowd-sourced Python programming problems, designed to be solvable by entry-level programmers, covering programming fundamentals, standard library functionality, and so on. Each problem consists of a task description, code solution and 3 automated test cases.

## C   IMPLEMENTATION DETAILS

We fine-tune all the SFT datasets with 3 epochs and a batch size of 16 on NVIDIA A100 GPUs. We use 8 GPUs for 7B and 13B models, 16 GPUs for 33B models during fine-tuning. We use a peak learning rate of 2e-5 with a 3% learning rate warmup. We evaluate the results on the final epoch. We use greedy decode to calculate Pass@1 and maj@1. Since the scores of MT-bench will fluctuate, we conducted three experiments and took the average.

All experiments are conducted using the default template of the FastChat framework (Zheng et al., 2023), as shown in the figure below:

> **Prompt Template**
>
> A chat between a curious user and an artificial intelligence assistant. The assistant gives helpful, detailed, and polite answers to the user's questions. **USER:** {Query} **ASSISTANT:**

## D   ESTIMATING FLOPs OF SFT

**Training FLOPs**   We mainly follow the notations of (Kaplan et al., 2020) here.

For each input sample of length $n_{ctx}$ in SFT dataset (GSM8K, CodeAlpaca, ShareGPT), we can split it into two parts:

$$n_{ctx} = n_Q + n_R \tag{1}$$

$$C_{\text{train}} \approx 6 N n_{ctx} N_s \tag{2}$$

where $n_Q, n_R$ denotes the length of question and generated answers respectively. $N, N_s$ denotes the non-embedding parameters and the numbers of samples.

Therefore, We estimate the SFT FLOPs following (Kaplan et al., 2020) and GPU times in Table 3.

## E   VALIDATION EXPERIMENTS IN MORE SFT ABILITIES

To validate the generalization of our conclusions, we selected representative datasets to evaluate the capabilities of large models across different dimensions. These dimensions include **World Knowledge** : WebQuestionsSP (Yih et al., 2016), **Language Understanding**: CoNLL 2003 (Tjong Kim Sang & De Meulder, 2003), and **Translation**: IWSLT14 (Cettolo et al., 2014)

**Experimental Design:**   Align the settings of RQ1 and RQ2, we introduce two settings as follows:

**1. Individual Domain:** We conduct SFT on LLaMA of various sizes using {1, 1/2, 1/4, 1/8} proportions [1] of the training set obtained from WebQSP, CoNLL 2003, and IWSLT14 seperately. This allowed us to evaluate each ability with various data sizes and model sizes.

**2. Mixed Domain:** We sampled {1, 1/2, 1/4, 1/8} amounts of training data from WebQSP, CoNLL 2003, and IWSLT14, and directly mixed them according to the corresponding proportions. In this way, we constructed datasets with fixed proportions of different ability domains, while varying the total data amount. These datasets are then used for fine-tuning the LLaMA models.

---

[1]Because these three datasets have relatively small amounts of data (a few thousand), the scaling range is from 100% of the data volume to 1/8 of the data volume.

| Model size | 7B | 13B | 33B |
|---|---|---|---|
| *GSM8k RFT* | | | |
| SFT FLOPs | $2.4 \times 10^{18}$ | $4.3 \times 10^{18}$ | $1.1 \times 10^{19}$ |
| SFT GPI hrs | 6.1 | 12.1 | 37.4 |
| *Code Alpaca* | | | |
| SFT FLOPs | $4.7 \times 10^{17}$ | $7.8 \times 10^{17}$ | $2.0 \times 10^{18}$ |
| SFT GPI hrs | 1.2 | 2.5 | 8.2 |
| *ShareGPT* | | | |
| SFT FLOPs | $2.2 \times 10^{18}$ | $3.9 \times 10^{18}$ | $9.7 \times 10^{19}$ |
| SFT GPI hrs | 5.4 | 10.9 | 34.0 |

Table 3: The statistics of FLOPs and GPU hours required for SFT. For 33B, we use DeepSpeed ZeRO3 (Rasley et al., 2020) for distributed training. All the GPU hours are based on NVIDIA A100 80GB GPU. Note we use non-embedding parameters to compute FLOPs in our experiments.

| Datasets | CONIL03 | | | WebQSP | | IWSLT14 | |
|---|---|---|---|---|---|---|---|
| | P | R | F1 | F1 | Hits@1 | de-en | en-de |
| Single Domain(1/1) | 91.89 | 89.33 | 90.59 | 33.5 | 64.12 | 50 | 52 |
| Single Domain(1/2) | 90.59 | 87.15 | 88.83 | 27.10 | 61.87 | 46 | 43 |
| Single Domain(1/4) | 85.24 | 79.46 | 82.25 | 22.56 | 61.38 | 42 | 40 |
| Single Domain(1/8) | 63.22 | 60.42 | 61.79 | 13.63 | 49.05 | 41 | 40 |
| Mixed Domains(1/1) | 91.74 | 87.79 | 89.72 | 32.10 | 63.70 | 46 | 49 |
| Mixed Domains(1/2) | 90.69 | 86.93 | 88.77 | 29.98 | 62.29 | 45 | 45 |
| Mixed Domains(1/4) | 88.81 | 85.62 | 87.18 | 25.42 | 58.02 | 43 | 43 |
| Mixed Domains(1/8) | 86.47 | 81.18 | 83.74 | 21.36 | 56.86 | 45 | 45 |

Table 4: Results in other domains for single and mixed source settings based on Llama-7B.

As shown in Table 4, we have following observations.

For the **individual domain**, the performance (P, R, F1) of the model in the language understanding (NER) task shows a positive correlation with the scaling curve of data volume. These two abilities exhibit similar scaling curve trends as the mathematical ability performance in RQ1. In the case of world knowledge (WebQSP), a similar positive correlation trend is observed in terms of F1 and Hits@1. However, when the data ratio is reduced from 1/4 to 1/8, there is a significant performance fluctuation, particularly in the performance of translation ability, which shows a relatively irregular trend. These conclusions further support the core conclusion of RQ1 that different data exhibit different scaling curves.

For the **mixed domains**, the findings align with the conclusions in RQ2, where abilities are improved with low-resource and decreased with high-resource compared to individual source abilities. This consistent conclusion holds for world knowledge, language understanding, and translation abilities.

# F    RESULTS ON MORE BENCHMARKS IN MATH AND CODE

To validate the generalization of our findings on other benchmarks, we utilized GSM8K and Code Alpaca as the training sets. We further evaluated the results on the individual domain, mixed domain,

| Methods | Math Benchmarks | | Code Benchmarks | |
|---|---|---|---|---|
| | GSM8K | MATH | HumanEval | MBPP |
| *Individual domain (Scaling)* | | | | |
| Single Domain(k=1/1) | 49.10 | 4.4 | 18.4 | 21.6 |
| Single Domain(k=1/4) | 43.37 | 3.9 | 11.58 | 18.8 |
| Single Domain(k=1/16) | 35.90 | 3.2 | 12.19 | 16.6 |
| Single Domain(k=1/64) | 22.71 | 3.2 | 9.14 | 15.8 |
| Single Domain(k=1/256) | 12.7 | 2.0 | 5.48 | 15.8 |
| *Mixed domain (Scaling)* | | | | |
| Mixed Domain(k=1/1) | 47.53 | 3.6 | 14.63 | 19.4 |
| Mixed Domain(k=1/4) | 41.98 | 3.2 | 9.14 | 20.6 |
| Mixed Domain(k=1/16) | 32.97 | 2.4 | 9.16 | 18.4 |
| Mixed Domain(k=1/64) | 25.77 | 2.4 | 14.63 | 17.2 |
| Mixed Domain(k=1/256) | 14.78 | 3.0 | 11.37 | 16.6 |
| *Individual domain* | | | | |
| General only | 11.1 | 2.9 | 10.4 | 1.0 |
| Math only | 49.10 | 4.4 | 6.71 | 9.0 |
| Code only | 4.51 | 1.0 | 18.40 | 21.6 |
| *Different Training Strategies* | | | | |
| Multi-task learning | 47.53 | 3.6 | 14.63 | 19.4 |
| Sequential Training | 31.39 | 2.0 | 15.85 | 15.8 |
| Mixed Sequential Training | 32.6 | 2.5 | 15.24 | 16.6 |
| DMT (k=1/256) | 41.92 | 3.4 | 17.68 | 18.8 |

Table 5: The detailed results of LLaMA-7B, 13B with different training strategies on three benchmarks.

and different training strategies on other specialized ability benchmark, including MATH and MBPP, which is illustrated in Table 5.

We have the following findings:

1. In the individual domain, Llama shows a positive correlation between performance in MATH and MBPP and the data volume (consistent with RQ1).

2. Comparing the individual and mixed domains, Llama-7B exhibits a trade-off between high-resource performance conflict and low-resource performance gain in both MATH and MBPP (consistent with RQ2).

3. Considering the general ability results shown in Table 1, we can observe that DMT maintains competitive results in MATH and MBPP while prioritizing general abilities. This further validates the effectiveness of DMT (consistent with RQ4).

## G  VISUALIZATION OF DIFFERENT LAYERS

In this section, we compared the visualization results of the baseline model of Llama-13B and DMT (k=1/256) in the starting layer (Layer1), middle layer (Layer15), and ending layer (Layer31) in Figure 7 and 8.

The visualization result of the starting layer are relatively chaotic, while the visualization results of the middle layer and the ending layer are clearer. And the results of the middle layer and the last layer are consistent in pointing out that both base model and model with DMT strategy exhibit a

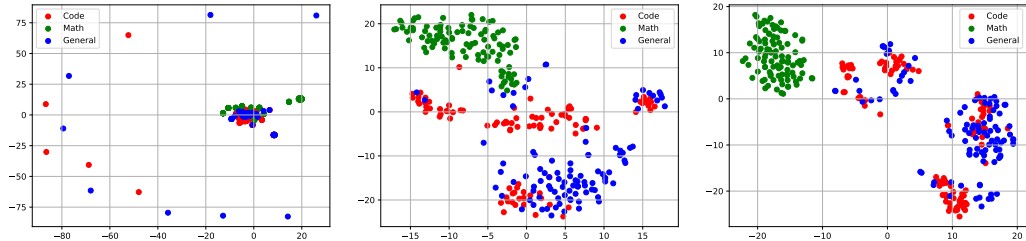

Figure 7: From left to right are the visualization results of starting layer (Layer1), middle layer (Layer15), and ending layer (Layer31) on Llama-7B.

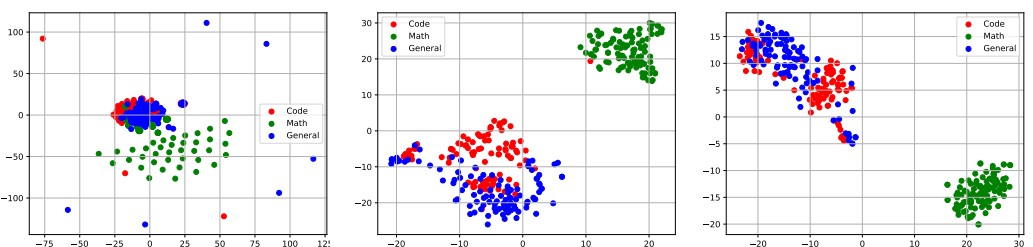

Figure 8: From left to right are the visualization results of starting layer (Layer1), middle layer (Layer15), and ending layer (Layer31) on Llama-7B with DMT(k=1/256) strategy.

certain level of separation in the mathematical data representations, there remains a certain degree of overlap between the representations of code and general samples.

## H  EQUAL DATA AMOUNT VS. EQUAL DATA PROPORTION

In a realistic SFT phrase for training general LLM, the data amount for different abilities is likely to differ. Therefore, instead of controlling the same amount of data, we select to mix datasets with the same proportion of subsets to better simulate real-world scenarios in all experiments. In addition, We further supplement the experimental results using different abilities mixed with the equal data amount and compare them with the results using the equal subset proportion in Table 6.

**Equal Data amount Setting:** we utilize the data amount of GSM8k RFT as the baseline. We sampled data with proportions of 1/16, 1/64, 1/256, and mixed samples of the same data amount from Code alpaca and ShareGPT.

**Equal Proportion Setting:** we sampled data with proportions of 1/16, 1/64, 1/256 according to the subset proportions of each dataset and mixed them, which is aligned with the setup in RQ2.

It can be observed that there is not a significant difference in the results of the three benchmark tests between the two settings. Therefore, these findings do not significantly impact the main experimental conclusions presented in the paper.

## I  COMPARISON EXPERIMENT OF DIFFERENT TRAINING SEQUENCES

Thank you for your suggestion. In this paper, we trained the models in the order of code → math → general abilities. However, to investigate the impact of training order on different SFT abilities, we

| Methods | GSM8K | HumanEval | MT-Bench |
|---|---|---|---|
| Mixed Domain(k=1/16, Equal Amount) | 34.49 | 9.14 | 5.49 |
| Mixed Domain(k=1/64, Equal Amount) | 25.02 | 13.54 | 5.21 |
| Mixed Domain(k=1/256, Equal Amount) | 16.7 | 11.54 | 4.63 |
| Mixed Domain(k=1/16, Equal Proportion) | 32.97 | 9.16 | 5.52 |
| Mixed Domain(k=1/64, Equal Proportion) | 25.77 | 14.63 | 5.24 |
| Mixed Domain(k=1/256, Equal Proportion) | 14.78 | 11.37 | 4.41 |

Table 6: Comparative experiment between equal data amounts and equal subset proportions of different SFT abilities on Llama-7B

| Methods | GSM8K | HumanEval | MT-Bench |
|---|---|---|---|
| Code $\rightarrow$ Math $\rightarrow$ General | 31.39 | 15.85 | **5.72** |
| Math $\rightarrow$ Code $\rightarrow$ General | 29.71 | 15.85 | **5.65** |
| Code $\rightarrow$ General $\rightarrow$ Math | **48.21** | 9.75 | 4.7 |
| General $\rightarrow$ Code $\rightarrow$ Math | **48.21** | 7.9 | 4.59 |
| General $\rightarrow$ Math $\rightarrow$ Code | 37.60 | **15.85** | 3.79 |
| Math $\rightarrow$ General $\rightarrow$ Code | 26.45 | **16.46** | 3.68 |

Table 7: Results of different sequential training for Llama-7B

have conducted additional experiments with six different training orders. The results and analysis of these experiments are provided in Table 7:

Based on our findings, we conclude the following:

1. The SFT ability trained in the final stage tend to retain relatively good performance.

2. If general and code abilities are trained in the first two stages, there is a noticeable performance decrease in code capability, while math capability does not show significant impact. One possible reason is that the task format of code generation and general ability exhibits similar data distributions (as discussed in RQ3 and Discussion1). This can result in a more severe catastrophic forgetting phenomenon during continuous fine-tuning.

# J   DETAILED RESULTS OF EXPERIMENTS

## J.1   RESULTS OF DIFFERENT RANDOM SEEDS

For each dataset, we employed random selection by utilizing a random function with three distinct seeds for sampling. Subsequently, we conducted a comparative analysis of the results obtained from different subsets on the three benchmark tests. The specific details are presented in Table 8. It can be observed that DMT maintains its superiority under three different random seed settings. The influence of different subsets on experimental results is not a key factor and does not affect the overall trend.

## J.2   RESULTS OF SINGLE SOURCE AND MIXED SOURCE

In Table 9 and Table 10, we report the detailed comparative results between mix domains and individual domains for LLaMA-7B, 3B and 33B, as the supplemental results in RQ2.

## J.3   RESULTS OF DATA RATIO (K)

In Table 11, we report The detailed results of the data ratio (k) between specific abilities and general abilities on three benchmarks, as the supplemental results in RQ3.

| Methods | LLaMA -7B | | | LLaMA -13B | | | LLaMA -33B | | |
|---|---|---|---|---|---|---|---|---|---|
| | GSM8K | HumanEval | MT-Bench | GSM8K | HumanEval | MT-Bench | GSM8K | HumanEval | MT-Bench |
| *Different Training Strategies* | | | | | | | | | |
| Multi-task learning | **47.53** | 14.63 | 5.76 | **50.94** | 19.50 | 5.73 | **56.69** | 18.9 | 6.07 |
| Sequential Training | 31.39 | 15.85 | 5.72 | 39.12 | 20.12 | 5.93 | 47.27 | 24.80 | **6.73** |
| Mixed Sequential Training | 32.60 | 15.24 | 6.02 | 40.48 | 18.30 | 5.93 | 44.24 | 24.4 | 6.43 |
| DMT(k=1/256,random seed=1) | 41.92 | 17.68 | 6.08 | 46.47 | 19.50 | 6.03 | 56.36 | 25.00 | 6.69 |
| DMT(k=1/256,random seed=2) | 41.31 | 17.68 | 6.02 | 45.85 | 18.90 | 6.08 | 55.64 | 24.80 | 6.71 |
| DMT(k=1/256,random seed=3) | 42.03 | **18.21** | **6.13** | 46.22 | **20.52** | **6.10** | 56.12 | **25.30** | **6.73** |

Table 8: The results of LLaMA-7B, 13B, 33B under different training strategies on three benchmarks. The top two results across different strategies are marked with **bold** and underlined. We tested the results of DMT on randomly sampling k proportion of specified data under three random seeds.

### J.4   RESULTS OF SPECIALIZED DATA AMOUNT OF DMT

In Table 12, we report The detailed results of LLaMA-7B, 13B, 33B with different training strategies on three benchmarks, as the supplemental results in RQ4.

### J.5   RESULTS OF MT-BENCH

In Figure 9, we report detailed results of LLaMA-7B, 13B, 33B with different training strategies on MT-Bench, which include coding, extraction, humanities, math, reasoning, roleplay, stem and writing abilities.

### J.6   SUPPLEMENTAL RESULTS FOR DICUSSION

In Figure 10, we report the t-SNE visualizations of LLaMA-7B and LLaMA-7B with DMT(k=1/256) strategy. What's more, the bottom figure represents the scaling relationship of LLaMA-7B with DMT(k=1/256) under different values of K.

Moreover, in Table 13, we report The detailed results of LLaMA-7B, 13B, 33B with different training strategies on three benchmarks, as the supplemental results in RQ4.

| Methods | LLaMA-7B | | | LLaMA-13B | | |
|---|---|---|---|---|---|---|
| | GSM8K | HumanEval | MT-Bench | GSM8K | HumanEval | MT-Bench |
| Single(k=1) | 49.10 | 18.4 | 5.88 | 51.4 | 18.4 | 6.13 |
| Single(k=1/4) | 43.37 | 11.58 | 5.85 | 48.59 | 13.41 | 6.03 |
| Single(k=1/16) | 35.90 | 12.19 | 5.61 | 43.00 | 12.80 | 5.66 |
| Single(k=1/64) | 22.71 | 9.14 | 5.11 | 27.40 | 12.20 | 5.24 |
| Single(k=1/256) | 12.70 | 5.48 | 4.00 | 18.40 | 10.36 | 2.95 |
| Mix(k=1) | 47.53 | 14.63 | 5.76 | 50.49 | 17.10 | 5.73 |
| Mix(k=1/4) | 41.98 | 9.14 | 5.48 | 48.52 | 14.00 | 5.61 |
| Mix(k=1/16) | 32.97 | 9.16 | 5.22 | 40.63 | 14.60 | 5.52 |
| Mix(k=1/64) | 25.77 | 14.63 | 5.27 | 33.2 | 17.68 | 5.24 |
| Mix(k=1/256) | 14.78 | 11.37 | 4.11 | 24.94 | 12.19 | 4.4 |

Table 9: Comparative experiments between mix domains and individual domains for LLaMA-7B, 13B.

| Methods | GSM8K | HumanEval | MT-Bench |
|---|---|---|---|
| Single(k=1) | 57.91 | 26.82 | 6.63 |
| Single(k=1/4) | 56.10 | 25.61 | 6.66 |
| Single(k=1/16) | 54.60 | 21.95 | 6.17 |
| Single(k=1/64) | 44.60 | 18.59 | 5.99 |
| Single(k=1/256) | 29.21 | 14.02 | 2.3 |
| Mix(k=1) | 56.69 | 18.9 | 6.07 |
| Mix(k=1/4) | 54.54 | 22.56 | 5.92 |
| Mix(k=1/16) | 53.33 | 26.82 | 6.26 |
| Mix(k=1/64) | 46.66 | 18.6 | 5.73 |
| Mix(k=1/256) | 36.54 | 17.68 | 4.58 |

Table 10: Comparative experiments between mix domains and individual domains for LLaMA-33B.

| Model size | GSM8K | HumanEval | MT-Bench |
|---|---|---|---|
| Mix[(code,math),1 general] | 47.53 | 14.63 | 5.76 |
| Mix[(code,math),1/4 general] | 48.44 | 15.85 | 5.73 |
| Mix[(code,math),1/16 general] | 47.99 | 15.24 | 5.27 |
| Mix[(code,math),1/64 general] | 47.23 | 14.63 | 5.16 |
| Mix[(code,math),1/256 general] | 48.52 | 16.46 | 4.69 |
| Mix[1(code,math),general] | 47.53 | 14.63 | 5.76 |
| Mix[1/4(code,math),general] | 41.31 | 10.97 | 5.81 |
| Mix[1/16(code,math),general] | 33.20 | 11.58 | 5.76 |
| Mix[1/64(code,math),general] | 25.17 | 12.19 | 5.84 |
| Mix[1/256(code,math),general] | 16.52 | 9.14 | 5.82 |
| Mix[1(code,math),1/64general] | 47.68 | 14.63 | 5.09 |
| Mix[1/4(code,math),1/64general] | 43.29 | 12.19 | 5.07 |
| Mix[1/16(code,math),1/64general] | 33.81 | 12.19 | 5.17 |
| Mix[1/64(code,math),1/64general] | 26.23 | 12.19 | 5.12 |
| Mix[1/256(code,math),1/64general] | 18.27 | 10.36 | 5.12 |

Table 11: The detailed results of the data ratio (k) between specific abilities and general abilities on three benchmarks.

| Methods | LLaMA-7B | | | LLaMA-13B | | |
|---|---|---|---|---|---|---|
| | GSM8K | HumanEval | MT-Bench | GSM8K | HumanEval | MT-Bench |
| *Individual domain* | | | | | | |
| General only | 11.1 | 10.4 | 5.88 | 14.02 | 16.4 | 6.13 |
| Math only | 49.1 | - | - | 51.4 | - | - |
| Code only | - | 18.4 | - | - | 17.1 | - |
| *Different Training Strategies* | | | | | | |
| Multi-task learning | 47.53 | 14.63 | 5.76 | 50.94 | 19.5 | 5.73 |
| Sequential Training | 31.39 | 15.85 | 5.72 | 39.12 | 20.12 | 5.93 |
| Mixed Sequential Training | 32.6 | 15.24 | 6.02 | 40.48 | 18.30 | 5.93 |
| DMT (k=1) | 45.79 | 14.02 | 5.63 | 50.49 | 16.46 | 5.76 |
| DMT (k=1/4) | 48.37 | 13.41 | 5.69 | 50.18 | 18.9 | 5.83 |
| DMT (k=1/16) | 43.3 | 15.24 | 5.78 | 48.59 | 18.9 | 5.96 |
| DMT (k=1/64) | 42.53 | 15.85 | 6.01 | 47.61 | 15.24 | 6.03 |
| DMT (k=1/256) | 41.92 | 17.68 | 6.08 | 46.47 | 19.5 | 6.03 |

Table 12: The detailed results of LLaMA-7B, 13B with different training strategies on three benchmarks.

| Model size | GSM8K | HumanEval | MT-Bench |
|---|---|---|---|
| 1/1 Mix(code,math,general(w/o code math)) | 49.05 | 17.68 | 5.80 |
| 1/4 Mix(code,math,general(w/o code math)) | 43.13 | 15.85 | 5.71 |
| 1/16 Mix(code,math,general(w/o code math)) | 36.23 | 10.36 | 5.38 |
| 1/64 Mix(code,math,general(w/o code math)) | 25.62 | 10.97 | 5.21 |
| 1/256 Mix(code,math,general(w/o code math)) | 15.31 | 11.37 | 4.38 |

Table 13: The scaling curve after ablating code and math-related samples from ShareGPT

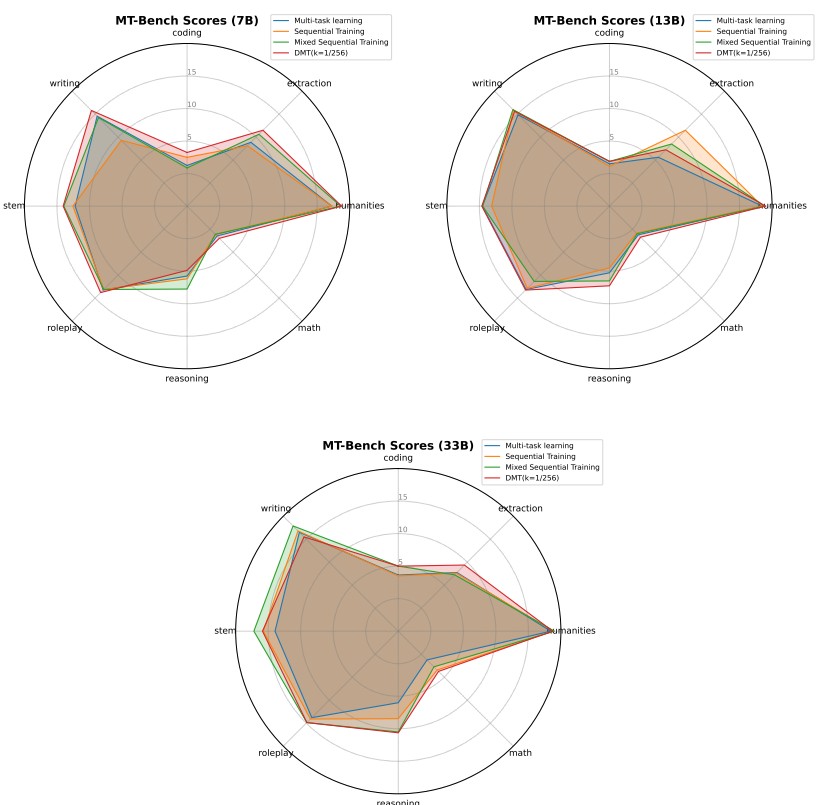

Figure 9: The detailed results of LLaMA-7B, 13B, 33B with different training strategies on MT-Bench.

| Methods | LLaMA -7B | | | LLaMA -13B | | | LLaMA -33B | | |
|---|---|---|---|---|---|---|---|---|---|
| | GSM8K | HumanEval | MT-Bench | GSM8K | HumanEval | MT-Bench | GSM8K | HumanEval | MT-Bench |
| *Individual domain* | | | | | | | | | |
| General only | 11.10 | 10.42 | 5.88 | 14.02 | 16.40 | 6.13 | 26.06 | 24.30 | 6.63 |
| Math only | 49.10 | 6.71 | 2.53 | 51.40 | 12.8 | 2.54 | 57.91 | 15.5 | 3.18 |
| Code only | 4.51 | 18.40 | 4.30 | 5.15 | 17.1 | 3.53 | 6.06 | 26.82 | 4.18 |
| *Different Training Strategies* | | | | | | | | | |
| Multi-task learning | **47.53** | 14.63 | 5.76 | **50.94** | 19.50 | 5.73 | **56.69** | 18.9 | 6.07 |
| Sequential Training | 31.39 | **15.85** | 5.72 | 39.12 | **20.12** | 5.93 | 47.27 | 24.80 | **6.73** |
| Mixed Sequential Training | 32.60 | 15.24 | 6.02 | 40.48 | 18.30 | 5.93 | 44.24 | 24.4 | 6.43 |
| DMT(k=1/256) | 41.92 | 17.68 | **6.08** | 46.47 | 19.50 | **6.03** | 56.36 | **25.00** | 6.69 |

Table 14: The results of LLaMA-7B, 13B, 33B under different training strategies on three benchmarks. The top two results across different strategies are marked with **bold** and underlined.

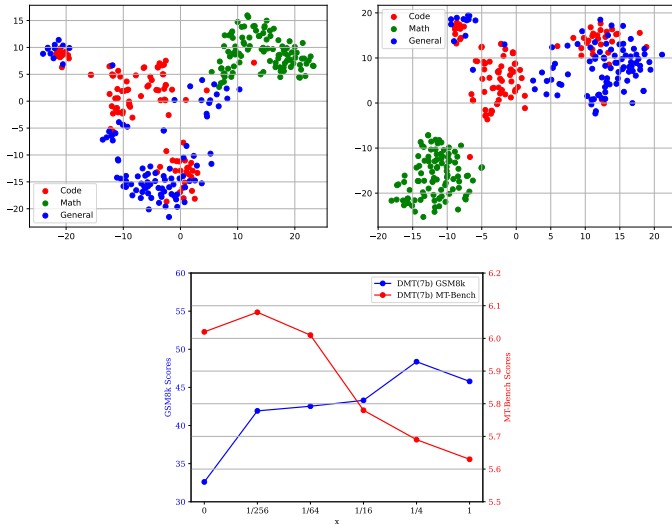

Figure 10: The upper two figures show the t-SNE visualizations of LLaMA-7B and LLaMA-7B with DMT(k=1/256) stategy. The bottom figure represents the scaling relationship of LLaMA-7B with DMT under different values of K.

| Methods | LLaMA -7B | | | LLaMA -13B | | | LLaMA -33B | | |
|---|---|---|---|---|---|---|---|---|---|
| | GSM8K | HumanEval | MT-Bench | GSM8K | HumanEval | MT-Bench | GSM8K | HumanEval | MT-Bench |
| *Individual domain* | | | | | | | | | |
| General only | 11.10 | 10.42 | 5.88 | 14.02 | 16.40 | 6.13 | 26.06 | 24.30 | 6.63 |
| Math only | 49.10 | 6.71 | 2.53 | 51.40 | 12.8 | 2.54 | 57.91 | 15.5 | 3.18 |
| Code only | 4.51 | 18.40 | 4.30 | 5.15 | 17.1 | 3.53 | 6.06 | 26.82 | 4.18 |
| *Different Training Strategies* | | | | | | | | | |
| Multi-task learning | **47.53** | 14.63 | 5.76 | **50.94** | 19.50 | 5.73 | **56.69** | 18.9 | 6.07 |
| Sequential Training | 31.39 | 15.85 | 5.72 | 39.12 | **20.12** | 5.93 | 47.27 | 24.80 | **6.73** |
| Mixed Sequential Training | 32.60 | 15.24 | 6.02 | 40.48 | 18.30 | 5.93 | 44.24 | 24.4 | 6.43 |
| DMT(k=1/256) | 41.92 | **17.68** | **6.08** | 46.47 | 19.50 | **6.03** | 56.36 | **25.00** | 6.69 |

Table 15: The results of LLaMA-7B, 13B, 33B under different training strategies on three benchmarks. The top two results across different strategies are marked with **bold** and underlined.

