# OpenReview forum: "How Abilities in Large Language Models are Affected by Supervised Fine-tuning Data Composition"
_ICLR.cc/2024/Conference — ICLR 2024 Conference Withdrawn Submission_

### Official Review · Reviewer_C1QY · 2023-10-29

**Soundness:** 3 good
**Presentation:** 3 good
**Contribution:** 3 good
**Rating:** 6
**Confidence:** 4

**Summary:**

- The paper investigates the impact of data composition on the coding, mathematical reasoning, and general instruction-following abilities of LLMs through supervised fine-tuning (SFT).
- It explores the scaling laws of these abilities concerning factors like data amounts, data composition ratios, model parameters, and SFT strategies. It also studies whether these abilities interact with each other.
- A new training strategy, Dual-stage Mixed Fine-tuning (DMT), is proposed to address the issue of ability conflicts and catastrophic forgetting during multi-task learning.

**Strengths:**

1. This is a good empirical paper with solid experimental results. It makes several important observations that might be helpful for later fine-tuning research. For instance, it is interesting to know that SFT data from different sources appears to benefit each other in low-resource settings, but may act as noise when sufficient data is available.

2. The presentation is clear and easy to follow.

3. Though there's less algorithmic novelty in this paper, the empirical problem it studies is important, and there doesn't seem to be previous work focusing on empirically understanding the compositionality and ability conflicts of LLMs.

**Weaknesses:**

1. Limited abilities/datasets: while the general problem of data composition and ability conflicts is interesting, the paper only focuses on three abilities---reasoning, coding, and aligning general human intentions---and uses only a single dataset for each ability. This raises the question of whether the observed trends and scaling laws can generalize.

2. Limited models: similarly, only the LLaMA families are evaluated. It would be better if the paper can test DMT on several other models so that we could know the empirical observations about training strategies can generalize.

3. The current submission does not contain the training and evaluation code. Hopefully this will be released to facilitate reproducibility when the paper is publicly available.

**Questions:**

1. General setup: What’s the zero-shot performance (i.e., 0 SFT data) of the models on the three datasets? It's good to have these results as a baseline.

2. RQ1: How is the data selection process performed for each training dataset? Is it random? If so, did you sample using multiple seeds and compared results of different subsets? Are they similar?

3. RQ2: Since the paper mixes the datasets using the same subset ratio instead of the same data amounts, it would be better to indicate the number of data samples for each dataset in the paper. For instance, if one dataset is extremely large and the others are small, will this affect learning the other abilities? Also, it would be interesting to see how the model performs when it is fine-tuned with the same number of data points from each training dataset (rather than same subset ratio).

4. RQ4: The paper discusses sequential training. What is the performance when training with math -> code -> general abilities? Is it different from code -> math -> general abilities?

5. Figure 5 uses the features from the 15th layer of the model. What is the rationale for selecting this particular layer? Would it be better to show results from the beginning or end of the layers?

6. The paper discusses the impact of the distribution gap on acquiring multiple abilities. In the last paragraph of section 4.2, it says that  "*This distribution gap introduces an extra noise during the SFT phrase, while its removal enables the model to better generalize coding and mathematical abilities.*" However, I don’t think better **test-time accuracy** means better generalization ability. It can be that the model learns better how to deal with (or even overfit to) the training distributions (after removing code & math from ShareGPT), and removing the data ensures that **the test distribution is similar to the training distribution**. But in reality, the model can encounter diverse data formats such as the ones removed, and we want to generalize to these diverse data formats rather than only being able to perform well on the data included in ShareGPT.

---

> ### Author Response · Authors · 2023-11-19
> **Rebuttal for Weakness 1**
>
> Thanks for your suggestions, we have further added more SFT abilities and more evaluation benchmarks to verify our generalizability.
> ### More abilities and datasets：
> We are willing to supplement the results of different domain datasets in **Appendix section titled "VALIDATION EXPERIMENTS IN MORE SFT ABILITIES"** to verify the generalizability of our conclusions. We have selected several representative datasets to expand the evaluation of the capabilities of LLMs in different dimensions, including **World knowledge (WebQuestionsSP), Language Understanding (CoNLL 2003), and Translation abilities (IWSLT14)**. The experimental results are as follows:
>
> | Datasets           |  CONIL03 (P / R / F1) | WebQSP (F1 / Hits@1) | IWSLT14 (de-en / en-de) |
> |--------------------|----------------------|----------------------|-------------------------|
> | Single Domain (1/1)| 91.89 / 89.33 / 90.59 | 33.5 / 64.12         | 50 / 52                 |
> | Single Domain (1/2)| 90.59 / 87.15 / 88.83 | 27.10 / 61.87        | 46 / 43                 |
> | Single Domain (1/4)| 85.24 / 79.46 / 82.25 | 22.56 / 61.38        | 42 / 40                 |
> | Single Domain (1/8)| 63.22 / 60.42 / 61.79 | 13.63 / 49.05        | 41 / 40                 |
> |--------------------|----------------------|----------------------|-------------------------|
> | Mixed Domains (1/1)| 91.74 / 87.79 / 89.72 | 32.10 / 63.70        | 46 / 49                 |
> | Mixed Domains (1/2)| 90.69 / 86.93 / 88.77 | 29.98 / 62.29        | 45 / 45                 |
> | Mixed Domains (1/4)| 88.81 / 85.62 / 87.18 | 25.42 / 58.02        | 43 / 43                 |
> | Mixed Domains (1/8)| 86.47 / 81.18 / 83.74 | 21.36 / 56.86        | 45 / 45                 |
>
>
> **Individual domain:** For the language understanding (NER) task, the performance (P, R, F1) of the model shows a positive correlation with the scaling curve of the data. These two abilities exhibit a similar performance scaling trend as the mathematical ability discussed in RQ1. On the other hand, for world knowledge (WebQSP) in terms of (F1, Hits@1), it also shows a positive correlation trend. However, when the data ratio decreases from 1/4 to 1/8, there is a significant fluctuation in performance. In the case of translation ability, the performance trends for German-English and English-German translations are not entirely consistent. These findings further support the core conclusion of RQ1 that different data exhibit different scaling curves.
>
> **Mixed domains:** The findings align with the conclusions of RQ2 in mixed doamins, which states that abilities are improved with low-resource and decreased with high-resource compared to individual source abilities. This consistent observation holds true for world knowledge, language understanding, and translation abilities.

---

> ### Author Response · Authors · 2023-11-19
> **Rebuttal of Weakness 2 and 3**
>
> ### Weakness2
>
> We will make every effort to supplement the results of more models before the end of the rebuttal period !
>
> ### Weakness3
>
> In our paper, all the models, datasets, evaluation benchmarks, and training frameworks used are open-source. We will provide explicit details of the experimental setup in the implementation section, ensuring transparency and reproducibility. Furthermore, we are committed to following our company's open-source procedures to make the code publicly available, advocating for increased accessibility and collaboration within the research community.

---

> ### Author Response · Authors · 2023-11-19
> **Rebuttal for Question 1 and 2**
>
> ### Question 1:
>
> Thanks for your suggestion, we have supplemented all zero-shot performance of Llama7B to 33B in 3 benchmarks in **rebuttal revised version in Table1** for reference.
> | Methods                     | LLaMA -7B (GSM8K / HumanEval / MT-Bench) | LLaMA -13B (GSM8K / HumanEval / MT-Bench) | LLaMA -33B (GSM8K / HumanEval / MT-Bench) |
> |-----------------------------|-----------------------------------------|------------------------------------------|------------------------------------------|
> | General only                | 11.10 / 10.42 / 5.88                     | 14.02 / 16.40 / 6.13                     | 26.06 / 24.30 / 6.63                     |
> | Math only                   | 49.10 / 6.71 / 2.53                      | 51.40 / 12.8 / 2.54                      | 57.91 / 15.5 / 3.18                      |
> | Code only                   | 4.51 / 18.40 / 4.30                      | 5.15 / 17.1 / 3.53                       | 6.06 / 26.82 / 4.18                      |
>
>
> ### Question 2:
>
> For each dataset, we employed random selection by utilizing a random function with three distinct seeds for sampling. Subsequently, we conducted a comparative analysis of the results obtained from different subsets on the three benchmark tests. The specific details are presented as follows:
>
> | Methods                     | LLaMA -7B (GSM8K / HumanEval / MT-Bench) | LLaMA -13B (GSM8K / HumanEval / MT-Bench) | LLaMA -33B (GSM8K / HumanEval / MT-Bench) |
> |-----------------------------|-----------------------------------------|------------------------------------------|------------------------------------------|
> | DMT(k=1/256,random seed=1)  | 41.92 / 17.68 / 6.08                      | 46.47 / 19.50 / 6.03                       | 56.36 / 25.00 / 6.69                      |
> | DMT(k=1/256,random seed=2)  | 41.31 / 17.68 / 6.02                      | 45.85 / 18.90 / 6.08                       | 55.64 / 24.80 / 6.71                      |
> | DMT(k=1/256,random seed=3)  | 42.03 / 18.21 / 6.13                      | 46.22 / 20.52 / 6.10                       | 56.12 / 25.30 / 6.73                      |
>
>
> It can be observed that DMT maintains its superiority under three different random seed settings. The influence of different subsets on experimental results is not a key factor and does not affect the overall trend. More results can be found in **Appendix section titled "RESULTS OF DIFFERENT RANDOM SEEDS"**

---

> ### Author Response · Authors · 2023-11-19
> **Rebuttal for Question 3**
>
> ### Data statistics:
> Thank you for your suggestion. In fact, we have indicated the total number of samples for each dataset in the **Appendix section titled "SFT Dataset"** . We will also provide a comparative table in this section, showcasing the data size for different datasets and their respective proportions.
> | Data Statistics | GSM8K RFT | Code Alpaca | ShareGPT |
> |-----------------|-----------|-------------|----------|
> | K=1/1           | 110142    | 20022       | 86060    |
> | K=1/4           | 27535     | 5005        | 21515    |
> | K=1/16          | 6883      | 1251        | 5378     |
> | K=1/64          | 1720      | 312         | 1344     |
> | K=1/256         | 430       | 78          | 336      |
>
>
>
> ### Equal Data Amount Experiment:
> In a realistic SFT phrase for training general LLM, the data amount for different abilities is likely to differ. Therefore, instead of controlling the same amount of data, we select to mix datasets with the same proportion of subsets to better simulate real-world scenarios in all experiments. In addition, We are willing to further supplement the experimental results using different abilities mixed with the equal data amount and compare them with the results using the equla subset proportion.
>
>
> | Methods                              | GSM8K | HumanEval | MT-Bench |
> |--------------------------------------|-------|-----------|----------|
> | Mixed Domain(k=1/16, Equal Amount)    | 34.49 | 9.14      | 5.49     |
> | Mixed Domain(k=1/64, Equal Amount)    | 25.02 | 13.54     | 5.21     |
> | Mixed Domain(k=1/256, Equal Amount)   | 16.7  | 11.54     | 4.63     |
> |--------------------------------------|-------|-----------|----------|
> | Mixed Domain(k=1/16, Equal Proportion)| 32.97 | 9.16      | 5.52     |
> | Mixed Domain(k=1/64, Equal Proportion)| 25.77 | 14.63     | 5.24     |
> | Mixed Domain(k=1/256, Equal Proportion)| 14.78 | 11.37     | 4.41     |
>
> **Equal Data amount Setting:** we utilize the data amount of GSM8k RFT as the baseline. We sampled data with proportions of {1/16, 1/64, 1/256}, and mixed samples of the same data amount from Code alpaca and ShareGPT.
>
> **Equal Proportion Setting:** we sampled data with proportions of {1/16, 1/64, 1/256} according to the subset proportions of each dataset and mixed them, which is aligned with the setup in RQ2.
>
> It can be observed that there is not a significant difference in the results of the three benchmark tests between the two settings. Therefore, these findings do not significantly impact the main experimental conclusions presented in the paper. Detailed results can be found in **Appendix section titled “EQUAL DATA AMOUNT VS. EQUAL DATA PROPORTION”.**

---

> ### Author Response · Authors · 2023-11-19
> **Rebuttal for Question 4 and 5**
>
> ### Question 4:
>
> Thank you for your suggestion. In this paper, we trained the models in the order of code -> math -> general abilities. However, to investigate the impact of training order on different SFT abilities, we have conducted additional experiments with six different training orders. The results and analysis of these experiments are provided below:
>
> | Methods                                   | GSM8K | HumanEval | MT-Bench |
> |-------------------------------------------|-------|-----------|----------|
> | Code -> Math -> General                   | 31.39 | 15.85     | 5.72     |
> | Math -> Code -> General                   | 29.71 | 15.85     | 5.65     |
> | Code -> General -> Math                   | 48.21 | 9.75      | 4.7      |
> | General -> Code -> Math                   | 48.21 | 7.9       | 4.59     |
> | General -> Math -> Code                   | 37.6  | 15.85     | 3.79     |
> | Math -> General -> Code                   | 26.45 | 16.46     | 3.68     |
>
>
> Based on our findings, we conclude the following:
>
> 1. The SFT ability trained in the final stage tend to retain relatively good performance.
>
> 2. If general and code abilities are trained in the first two stages, there is a noticeable performance decrease in code capability, while math capability does not show significant impact. One possible reason is that the task format of code generation and general ability exhibits similar data distributions (as discussed in RQ3 and Discussion1). This can result in a more severe catastrophic forgetting phenomenon during continuous fine-tuning.
>
> Detailed results can be found in **Appendix section titled “COMPARISON EXPERIMENT OF DIFFERENT TRAINING SEQUENCES”.**
>
> ### Question 5:
>
> Based on your suggestions, we have also added the comparison results between the start layer and the end layer in the **Appendix section titled “VISUALIZATION OF DIFFERENT LAYERS".** The visualization result of the start layer are relatively chaotic, while the visualization results of the middle layer and the end layer are clearer. And the results of the middle layer and the last layer are consistent in pointing out that both base model and model with DMT strategy exhibit a certain level of separation in the mathematical data representations, there remains a certain degree of overlap between the representations of code and general samples.
>
> For relevant support, please refer to Figure 18 in Section 6 in this paper [1]. It is also found that the Middle layer has a more stable visualization effect compared to the early layer.
>
> [1] REPRESENTATION ENGINEERING: A TOP-DOWN APPROACH TO AI TRANSPARENCY

---

> ### Author Response · Authors · 2023-11-19
> **Rebuttal for Question 6**
>
> Thank you for your suggestion. We have added the results of the mathematics benchmark MATH and the code benchmark MBPP in the Appendix "RESULTS ON MORE BENCHMARKS IN MATH AND CODE". Their test data belongs to the same field as GSM8k and Code Alpaca, but the distribution is relatively Out-of-distribution. It can be seen from the above results that our DMT strategy does have a certain degree of generalization. On the premise of ensuring excellent general ability performance, it has good results in the IND and OOD benchmarks of mathematics and code. The following are detailed results
>
>
> | Methods                     | MATH                           | MBPP                            |
> |-----------------------------|--------------------------------|---------------------------------|
> | Single Domain (k=1/1)       | 4.4                            | 21.6                            |
> | Single Domain (k=1/4)       | 3.9                            | 18.8                            |
> | Single Domain (k=1/16)      | 3.2                            | 16.6                            |
> | Single Domain (k=1/64)      | 3.2                            | 15.8                            |
> | Single Domain (k=1/256)     | 2.0                            | 15.8                            |
> |-----------------------------|--------------------------------|---------------------------------|
> | Mixed domain (Scaling)      |                                |                                 |
> | Mixed Domain (k=1/1)        | 3.6                            | 19.4                            |
> | Mixed Domain (k=1/4)        | 3.2                            | 20.6                            |
> | Mixed Domain (k=1/16)       | 2.4                            | 18.4                            |
> | Mixed Domain (k=1/64)       | 2.4                            | 17.2                            |
> | Mixed Domain (k=1/256)      | 3.0                            | 16.6                            |
> |-----------------------------|--------------------------------|---------------------------------|
>  | General only|  2.9 |  1.0 |
> | Math only                   | 4.4                            | 9.0                             |
> | Code only                   | 1.0                            | 21.6                            |
> |--------------------|----------------------|----------------------|-------------------------|
> | Multi-task learning         | 3.6                            | 19.4                            |
> | Sequential Training         | 2.0                            | 15.8                            |
> | Mixed Sequential Training   | 2.5                            | 16.6                            |
> | DMT (k=1/256)               | 3.4                            | 18.8                            |
>
>
> We have made the following findings:
> 1. In individual domains, the performance of the Llama model in MATH and MBPP is positively correlated with the data volume, similar to the findings in RQ1.
> 2. When comparing individual and mixed domains, the llama-7B model exhibits the same pattern of conflicting high-resource performance and low-resource performance gains in MATH and MBPP, consistent with the findings in RQ2.
> 3. Combining the results of general abilities in Table 1, we observe that DMT maintains competitive results in MATH and MBPP while prioritizing general abilities. This further validates the effectiveness of DMT, consistent with the findings in RQ4.
>
> The DMT strategy does have a certain degree of generalization. On the premise of ensuring excellent performance in general capabilities, it has good results in the IND and OOD benchmarks of mathematics and code. More detailed results we have added in **Appendix section titled “RESULTS ON MORE BENCHMARKS IN MATH AND CODE”**

---

> > ### Comment · Reviewer_C1QY · 2023-11-22
> >
> > I appreciate the authors' response and my concerns are mostly addressed. I'm happy to keep my rating.

---

### Official Review · Reviewer_crg6 · 2023-11-01

**Soundness:** 2 fair
**Presentation:** 3 good
**Contribution:** 2 fair
**Rating:** 5
**Confidence:** 3

**Summary:**

This paper studies the impact of different training strategies for supervised fine-tuning of language models for downstream tasks.  Further experiments are conducted to investigate how data composition, amount of data, and data composition ratios impact downstream performance.  Performance is evaluated on a subset of tasks for LLMs including code, symbolic (math) knowledge, and general human alignment.

**Strengths:**

* This paper presents a thorough evaluation of dataset composition, data size, and data composition ratios on different training strategies for supervised fine-tuning of LLMs.   Based on this, the authors propose a new fine-tuning strategy which is in-turn well-motivated.

* The authors find some surprising trends in the analysis namely that scaling data amount and data ratios on code in mixed settings have unreliable effects, whereas for other tasks this is not the case.  See Q2 regarding code mixing for training.

**Weaknesses:**

* All evaluations are done only on a single dataset from chosen domains: math, code, general human,  However, this does not cover different datasets within these domains or across other evaluations including world knowledge, language understanding,  reasoning, fairness, etc.  Many of the claims that are made in the paper such as "different abilities exhibit different scaling curves" and "larger models show better performance with the same data amount generally" feel too strong given the limited evaluations which may be anecdotal to the datasets and specific tasks chosen.  Many of the inconsistency results could also be attributed to differences in the dataset size or the data the model was trained on.  I would like to see more tasks and another dataset comparison before making these large claims as there are many other inconsistencies between the tasks that could lead to differing trends and performance differences beyond just the tasks themselves.

* Results in Table 1 are missing comparisons between datasets (fine-tuning on math and zero-shot or eval on HumanEval).  Results appear to be not much better for DMT compared with existing approaches, and DMT appears to be very particular to these tasks/datasets.  The paper would be much stronger with evaluation on another set of tasks/datasets to show the strategy will generalize.

* The section on scaling laws in LLMs seems a bit out of place in the related work as computing scaling laws is not a primary focus of this work, and there are no conclusions on the amount of data needed based on model size or compute budget for fine-tuning.

**Questions:**

Q1: There appears to be no reliable trend in the results with DMT and varying k on each of the datasets in Table 5.  What is the intuition for how to select k, and why there is inconsistency.  Intuitively more data should always help.

Q2: Results on code mixing having negative effects with more data seemed to be at odds with some of the results presented in prior work: https://arxiv.org/abs/2305.16264 which showed that adding code could actually help with NL tasks by adding structure.  Clarification on the results and why they're expected is beneficial.

---

> ### Author Response · Authors · 2023-11-19
> **Rebuttal for Weakness1**
>
> Thank you for your suggestions. In fact, as mentioned in the first paragraph of our introduction, our paper primarily focuses on three outstanding abilities (mathematics, coding, and general human-aligned ability) that are of great interest to the LLM community. In addition,  we are willing to supplement the results of different domain datasets in **Appendix section titled "VALIDATION EXPERIMENTS IN MORE SFT ABILITIES"** to verify the generalizability of our conclusions. We have selected several representative datasets to expand the evaluation of the capabilities of LLMs in different dimensions, including **World knowledge (WebQuestionsSP [1]), Language Understanding (CoNLL 2003 [2]), and Translation abilities (IWSLT14 [3])**. The experimental results are as follows:
>
> | Datasets           |  CONIL03 (P / R / F1) | WebQSP (F1 / Hits@1) | IWSLT14 (de-en / en-de) |
> |--------------------|----------------------|----------------------|-------------------------|
> | Single Domain (1/1)| 91.89 / 89.33 / 90.59 | 33.5 / 64.12         | 50 / 52                 |
> | Single Domain (1/2)| 90.59 / 87.15 / 88.83 | 27.10 / 61.87        | 46 / 43                 |
> | Single Domain (1/4)| 85.24 / 79.46 / 82.25 | 22.56 / 61.38        | 42 / 40                 |
> | Single Domain (1/8)| 63.22 / 60.42 / 61.79 | 13.63 / 49.05        | 41 / 40                 |
> |--------------------|----------------------|----------------------|-------------------------|
> | Mixed Domains (1/1)| 91.74 / 87.79 / 89.72 | 32.10 / 63.70        | 46 / 49                 |
> | Mixed Domains (1/2)| 90.69 / 86.93 / 88.77 | 29.98 / 62.29        | 45 / 45                 |
> | Mixed Domains (1/4)| 88.81 / 85.62 / 87.18 | 25.42 / 58.02        | 43 / 43                 |
> | Mixed Domains (1/8)| 86.47 / 81.18 / 83.74 | 21.36 / 56.86        | 45 / 45                 |
>
> **Individual domain**：For the language understanding (NER) task, the performance (P, R, F1) of the model shows a positive correlation with the scaling curve of the data. These two abilities exhibit a similar performance scaling trend as the mathematical ability discussed in RQ1. On the other hand, for world knowledge (WebQSP), it also shows a positive correlation trend. However, when the data ratio decreases from 1/4 to 1/8, there is a significant fluctuation in performance. In the case of translation, the performance trends for German-English and English-German translations are not entirely consistent. These findings further support the core conclusion of RQ1 that different data exhibit different scaling curves.
>
> **Mixed domains**: The findings align with the conclusions of RQ2 in mixed domains, which states that abilities are improved with low-resource and decreased with high-resource compared to individual source abilities.
>
> We further evaluated the bias of the models by assessing Truthful QA. We used the sharegpt data as the training set for SFT and explored the experimental setup of scaling the data from 100% to 1/256. We aligned the experimental setup with the evaluation criteria of TruthfulQA [4][5].
>
> | Methods                          | TruthfulQA (Standard Zero-shot) | TruthfulQA (w. truthful Prompt) |
> |----------------------------------|---------------------------------|---------------------------------|
> | Single Domain (1/1)              | 23.6                            | 26.5                            |
> | Single Domain (1/4)              | 22.76                           | 26.19                           |
> | Single Domain (1/16)             | 24.96                           | 26.43                           |
> | Single Domain (1/64)             | 22.03                           | 25.43                           |
> | Single Domain (1/256)            | 23.6                            | 26.5                            |
>
> The results indicate that the performance of lama-7B on TruthfulQA  did not show significant fluctuations. One possible reason for this is that the general ability data (sharegpt) may not enhance the abillity of debias. This will be a direction for future exploration.
>
> We will make appropriate adjustments to the conclusion statements in my article, avoiding controversial points to prevent any misinterpretation of the claims. We will also continue to explore whether the current conclusions hold true in other domains in the future.
>
> 【1】WebQuestionsSP：A question answering dataset using Freebase as the knowledge base for qustion answering.
>
> 【2】CoNLL 2003：CoNLL-2003 is a named entity recognition dataset released as a part of CoNLL-2003 shared task: language-independent named entity recognition.
>
> 【3】IWSLT14: Evaluation Campaign featured three tracks: automatic speech recognition (ASR), spoken language translation (SLT), and machine translation (MT).
>
> 【4】Representation Engineering: A Top-Down Approach to AI Transparency
>
> 【5】Investigating Uncertainty Calibration of Aligned Language Models under the Multiple-Choice Setting

---

> ### Author Response · Authors · 2023-11-19
> **Rebuttal for Weakness2**
>
> Thank you for your suggestion. We have added the following experimental results：
>
> **Zeroshot results**
>
> In the revised version of Table 1, we have provided the zero-shot results of single domain SFT models on other datasets.
>
> | Methods                     | LLaMA -7B (GSM8K / HumanEval / MT-Bench) | LLaMA -13B (GSM8K / HumanEval / MT-Bench) | LLaMA -33B (GSM8K / HumanEval / MT-Bench) |
> |-----------------------------|-----------------------------------------|------------------------------------------|------------------------------------------|
> | General only                | 11.10 / 10.42 / 5.88                     | 14.02 / 16.40 / 6.13                     | 26.06 / 24.30 / 6.63                     |
> | Math only                   | 49.10 / 6.71 / 2.53                      | 51.40 / 12.8 / 2.54                      | 57.91 / 15.5 / 3.18                      |
> | Code only                   | 4.51 / 18.40 / 4.30                      | 5.15 / 17.1 / 3.53                       | 6.06 / 26.82 / 4.18                      |
>
>
> **Results on more benchmarks**
>
> To validate the generalization of our findings on other benchmarks, we utilized GSM8K and Code Alpaca as the training sets. We further evaluated the results on the individual domain, mixed domain, and different training strategies on other specialized ability benchmark, including MATH and MBPP
>
> | Methods                     | MATH                           | MBPP                            |
> |-----------------------------|--------------------------------|---------------------------------|
> | Single Domain (k=1/1)       | 4.4                            | 21.6                            |
> | Single Domain (k=1/4)       | 3.9                            | 18.8                            |
> | Single Domain (k=1/16)      | 3.2                            | 16.6                            |
> | Single Domain (k=1/64)      | 3.2                            | 15.8                            |
> | Single Domain (k=1/256)     | 2.0                            | 15.8                            |
> |-----------------------------|--------------------------------|---------------------------------|
> | Mixed domain (Scaling)      |                                |                                 |
> | Mixed Domain (k=1/1)        | 3.6                            | 19.4                            |
> | Mixed Domain (k=1/4)        | 3.2                            | 20.6                            |
> | Mixed Domain (k=1/16)       | 2.4                            | 18.4                            |
> | Mixed Domain (k=1/64)       | 2.4                            | 17.2                            |
> | Mixed Domain (k=1/256)      | 3.0                            | 16.6                            |
> |-----------------------------|--------------------------------|---------------------------------|
>  | General only|  2.9 |  1.0 |
> | Math only                   | 4.4                            | 9.0                             |
> | Code only                   | 1.0                            | 21.6                            |
> |--------------------|----------------------|----------------------|-------------------------|
> | Multi-task learning         | 3.6                            | 19.4                            |
> | Sequential Training         | 2.0                            | 15.8                            |
> | Mixed Sequential Training   | 2.5                            | 16.6                            |
> | DMT (k=1/256)               | 3.4                            | 18.8                            |
>
>
> We have made the following findings:
>
> 1. In individual domains, the performance of the Llama model in MATH and MBPP is positively correlated with the data volume, similar to the findings in RQ1.
>
> 2. When comparing individual and mixed domains, the llama-7B model exhibits the same pattern of conflicting high-resource performance and low-resource performance gains in MATH and MBPP, consistent with the findings in RQ2.
>
> 3. Combining the results of general abilities in Table 1, we observe that DMT maintains competitive results in MATH and MBPP while prioritizing general abilities. This further validates the effectiveness of DMT, consistent with the findings in RQ4.
>
> More detailed results we have added in **Appendix section titled “RESULTS ON MORE BENCHMARKS IN MATH AND CODE”.**

---

> ### Author Response · Authors · 2023-11-19
> **Rebuttal for Weakness3**
>
> Thank you for your suggestion. We will incorporate the relevant literature on scaling laws into the appendix and include the calculation process for FLOPs，Training FLOPs budgets and GPU hours required for different datasets in the **Appendix section titled "ESTIMATING FLOPS OF SFT".**
>
> | Model size  | 7B             | 13B            | 33B             |
> |-------------|----------------|----------------|-----------------|
> | SFT FLOPs （GSM8k）   | 2.4e18         | 4.3e18         | 1.1e19          |
> | SFT GPI hrs  （GSM8k）| 6.1            | 12.1           | 37.4            |
> | SFT FLOPs （ Code Alpaca）  | 4.7e17         | 7.8e17         | 2.0e18          |
> | SFT GPI hrs （ Code Alpaca）| 1.2            | 2.5            | 8.2             |
> | SFT FLOPs （ ShareGPT）  | 2.2e18         | 3.9e18         | 9.7e19          |
> | SFT GPI hrs （ ShareGPT） | 5.4            | 10.9           | 34.0            |

---

> ### Author Response · Authors · 2023-11-19
> **Rebuttal for Question 1 and 2**
>
> **Q1:** The trend of k is indeed reliable. The inconsistency you mentioned seems to be referring to the increase of k from 0 to 1/256, which aligns with the conclusion of RQ2. As k increases from 0 to 1/256, a small amount of specialized data is mixed into the general ability data. This leads to a mutual benefit between general abilities and extremely low-resource specialized abilities, consistent with the conclusion of mutual benefit among different ability items under low-resource conditions in RQ2.
> However, as more specialized data is gradually introduced (k increasing from 1/256 to 1/1), the specialized abilities improve, but there is a gradual emergence of conflicting high-resource performance among different ability items (as observed in RQ2), ultimately resulting in a decline in general abilities.
>
>
>
> **Q2:** We acknowledge that incorporating code can enhance the generalization ability in other domains, as seen in the previous work. However, the key difference lies in the amount of code data used for SFT. In those studies, SFT was performed using code data with token counts exceeding 10 billion, allowing the models to capture transferable structured knowledge. In contrast, the token count of code data in our experiments is even less than 0.01 billion, which makes it challenging for large models to learn transferable knowledge. This significant difference in experimental setup accounts for the contrasting conclusions.
>
> In our experiments, we find that the factors influencing code mixing are more related to the style, format, and distribution of inputs from different domains (RQ3). We also acknowledge that investigating the effects of large-scale data with diverse abilities will be a valuable direction for future research.

---

> > ### Comment · Reviewer_crg6 · 2023-11-22
> > **Response to author comments**
> >
> > I thank the authors for providing several additional experiments on additional benchmark tasks.  The experiments addressing Weakness 1 improve the generalizability of the method.  However, the results still highlight that results primarily only improve in the low-resource settings indicating strong limitations of the proposed approach.  I will increase the score but am still not in favor of acceptance given the limited improvements.

---

### Official Review · Reviewer_4phB · 2023-11-01

**Soundness:** 2 fair
**Presentation:** 2 fair
**Contribution:** 1 poor
**Rating:** 3
**Confidence:** 4

**Summary:**

The manuscript studies data composition's effects on supervised fine-tuning in language modeling, especially large language models. The work proposes a new strategy for dataset mixing, incorporated into the pipeline of LLaMA, that demonstrates competitive behavior on general and specialized benchmarks like MT-Bench and (GSM8k, HumanEval) respectively.

**Strengths:**

1. Straightforward methodology
2. Experimental results are extensive in the aspects that the author care about
3. Experiments are performed on open models, which make the process replicable.

**Weaknesses:**

1. The analyses are very superficial, as only trends in standard benchmarks are measured and discussed. It is unclear what significance do t-SNE graphs show. The authors did not attempt to arrive at any form of empirical laws for typical scaling law works.
2. The main strategy has no novelty, as it's simply a mixture of different datasets. It seems very straightforward that adjusting the mixture will result in different capabilities on benchmarks, as well as the general rule of thumb that "more data under proper learning set-up results in better performance." The authors did not attempt to give any theoretical analysis of what they have observed, which further weakens the novelty claim.
3. Some wordings in the manuscript are unclear and unfit for an academic context. e.g. data amount, 100 thousand samples.

**Questions:**

What does it even mean for "scaling" of SFT when it deals with fractional data? I'm failing to see why the problem the author is studying is significant. Typical instruction fine-tuning works deal with a fine-tuning budget of 1E18 - 5E21 FLOPs [1], how would that translate to the amount of compute used in this work (since it's not clearly indicated)? Suppose the actual range of compute budget being studied (e.g. 1/256 "data amount" to 1 "data amount,") is much less than the usual amount in the literature. How would one ascertain that the scaling capabilities would extrapolate?

[1]: Scaling Instruction-Finetuned Language Models

**Details Of Ethics Concerns:**

This is not a grave concern, but DMT is simply, in my opinion, not a great name for research work due to the contraction's other denotations (https://en.wikipedia.org/wiki/N,N-Dimethyltryptamine, https://www.ncbi.nlm.nih.gov/pmc/articles/PMC6088236/).

---

> ### Author Response · Authors · 2023-11-16
>
> Thanks for your insightful comments.
>
> W1: T-SNE graphs try to understand the task distribution and we find math reasoning dataset is very different from others and ShareGPT does have overlap with coding tasks which helps us understand the performance change under multi-task learning. We don't give a scaling law equation since we do not use the test losses to evaluate task performance. Chinchilla's law using losses versus model size and data amount cannot be easily transformed here since metric like MT-Bench is not a simple function can be determined by the test losses, it is unknown that the power law still exists using metric like MT-Bench. However, MT-Bench/Alpaca-Eval are useful for quickly understanding human alignment performance and should be researched.
>
> W2: Our core finding is during multi-task learning, multiple abilities exhibits improvement in low-resource and decline in high-resource. Using our method can relief the decline in high-resource. Our main method should be defined as training routine instead of data mixture which trains on specialized tasks first and multi-task trains on human alignment data with a little specialized tasks. This method is emprically useful for LLM practioner to build a human-aligned chatbot with specialized abilities.
>
> W3: Thank you for your suggestion, we will improve our written and update the manuscript later.
>
> Q1: Our experiment setting is focus on aligning a pre-trained model to a chat model which mostly requires training samples from 1000 (less is more alignment) to 100,000 (ShareGPT). So our investigated fine-tuning budget is proper for our object.
>
> Ethics Concern: Thank you for your suggestion, we will change our method name.

---

### Author Response · Authors · 2023-11-19
**General Response**

Dear Reviewers:

First and foremost, we would like to express our sincere gratitude for reviewing our paper and providing valuable insights and suggestions. We appreciate that the reviewer's praise of our work's **strong motivation** (crg6, C1QY), **simple and efficient approach** (4phB, crg6), **extensive and solid experiments** (4phB, C1QY), **presentation clear and replicable** (4phB, crg6, C1QY).

As a comprehensive empirical study for the data composition of the SFT phrase, we summarize our core contributions here:

1. Different SFT abilities exhibit distinct scaling patterns, while larger models show better performances with the same data amount generally.

2. Mixed SFT abilities for finet-uning exhibits improvement in low-resource and decline in high-resource. As the model size increases, there is a greater performance gain in low-resource settings for math and general abilities.

3. Data amounts directly influence each ability, while the data ratio is insignificant

4. Multi-task learning lead to conflicts, while sequential training results in catastrophic forgetting. Our proposed DMT effectively alleviates both performance conflicts and catastrophic forgetting in the SFT phrase, achieving a balance between general and specialized abilities

To improve the paper's quality, we respond to the reviewers’ comments by making the following revisions (marked in red) to the paper:

1. **RQ4: Performance Difference vs. Training Strategy (Revised):** We have included the zero-shot results for each capability in Table 1.

2. **Appendix A: SFT Datasets (Revised):** We have provided data statistics to showcase the data size for different datasets and their respective proportions.

3. **Appendix B: Evaluation Metrics (Revised):** We have supplemented additional benchmarks, namely MBPP and MATH

4. **Appendix D: Estimating FLOPs of SFT (New):** We have provided the calculation process for FLOPs, training FLOPs budgets, and GPU hours required for different datasets in SFT.

5. **Appendix E: Validation Experiments in More SFT Abilities (New):** We have expanded the evaluation of LLMs' capabilities in different dimensions, including world knowledge, language understanding, and translation abilities.

6. **Appendix F: Results on More Benchmarks in Math and Code (New):** We have included additional benchmarks that exhibit relative distribution out-of-distribution (OOD) to verify the generalization of our conclusions.

7. **Appendix G: Visualization of Different Layers (New):** We have added a comparison of start and end layers using TSNE visualization.

8. **Appendix H: Equal Data Amount vs. Equal Data Proportion (New):** We have compared the performance of different capability data mixed with an equal data amount and equal data proportion.

9. **Appendix I: Comparison Experiment of Different Training Sequences (New):** We have investigated the impact of different capability sequences on performance in the ablation study of training sequences.

10. **Appendix J.1: Results of Different Random Seeds (New):** We have examined the influence of different random seeds on performance when sampling data with different proportions (k).

We believe that these modifications and additions better address your concerns and enhance the quality and readability of the paper. Once again, we sincerely appreciate your review, and we hope you will reconsider the review scores assigned to our work after reviewing this rebuttal.

Best regards.